DISCOVERY REPORT

# Fungal and host protein persulfidation are functionally correlated and modulate both virulence and antifungal response

**Monica Sueiro-Olivares[1], Jennifer Scott[1], Sara Gago[1], Dunja Petrovic[2,3], Emilia Kouroussis[2,3], Jasmina Zivanovic[2,3], Yidong Yu[4], Marlene Strobel[4], Cristina Cunha[5,6], Darren Thomson[1], Rachael Fortune-Grant[1], Sina Thusek[4], Paul Bowyer[1], Andreas Beilhack[4], Agostinho Carvalho[5,6], Elaine Bignell[1¤a], Milos R. Filipovic[7], Jorge Amich[1] \***

1 Manchester Fungal Infection Group (MFIG), School of Biological Sciences, Faculty of Biology, Medicine and Health, University of Manchester, Manchester Academic Health Science Centre, Manchester, United Kingdom, 2 Centre National de la Recherche Scientifique (CNRS), Institut de Biochimie et Genetique Cellulaires (IBGC), Bordeaux, France, 3 Université de Bordeaux, Institut de Biochimie et Genetique Cellulaires (IBGC), Bordeaux, France, 4 Interdisciplinary Center for Clinical Research (IZKF) Laboratory for Experimental Stem Cell Transplantation, Department of Internal Medicine II, University Hospital, Würzburg, Germany, 5 Life and Health Sciences Research Institute (ICVS), School of Medicine, University of Minho, Braga, Portugal, 6 Life and Health Sciences Research Institute (ICVS)/Biomaterials, Biodegradables and Biomimetics (3B's)—PT Government Associate Laboratory, Guimarães, Braga, Portugal, 7 Leibniz Institute for Analytical Sciences (ISAS), Dortmund, Germany

¤a Current address: Medical Research Council Centre for Medical Mycology, University of Exeter, United Kingdom

\* jorge.amichelias@manchester.ac.uk

The Editors encourage authors to publish research updates to this article type. Please follow the link in the citation below to view any related articles.

## Abstract

*Aspergillus fumigatus* is a human fungal pathogen that can cause devastating pulmonary infections, termed "aspergilloses," in individuals suffering immune imbalances or underlying lung conditions. As rapid adaptation to stress is crucial for the outcome of the host–pathogen interplay, here we investigated the role of the versatile posttranslational modification (PTM) persulfidation for both fungal virulence and antifungal host defense. We show that an *A. fumigatus* mutant with low persulfidation levels is more susceptible to host-mediated killing and displays reduced virulence in murine models of infection. Additionally, we found that a single nucleotide polymorphism (SNP) in the human gene encoding cystathionine γ-lyase (CTH) causes a reduction in cellular persulfidation and correlates with a predisposition of hematopoietic stem cell transplant recipients to invasive pulmonary aspergillosis (IPA), as correct levels of persulfidation are required for optimal antifungal activity of recipients' lung resident host cells. Importantly, the levels of host persulfidation determine the levels of fungal persulfidation, ultimately reflecting a host–pathogen functional correlation and highlighting a potential new therapeutic target for the treatment of aspergillosis.

**Data Availability Statement:** All raw data that support the findings of this study is available in the supplementary figures, tables and datasets.

**Funding:** JA was supported by a Medical Research Council (MRC, https://mrc.ukri.org/) Career Development Award (MR/N008707/1). SG was co-funded by the National Institute for Health Research (NIHR) Manchester Biomedical Research Centre (https://www.manchesterbrc.nihr.ac.uk/) and a National Centre for the Replacement, Refinement & Reduction of Animals in research (NC3Rs, https://www.nc3rs.org.uk/) Training Fellowship (NC/P002390/1). CC and AG were supported by the Northern Portugal Regional Operational Programme (NORTE 2020, https://ec. europa.eu/growth/tools-databases/regional-innovation-monitor/policy-document/norte-2020-norte%E2%80%99s-regional-operational-programme-2014-2020), under the Portugal 2020 Partnership Agreement, through the European Regional Development Fund (FEDER) (NORTE-01-0145-FEDER-000013), and the Fundação para a Ciência e Tecnologia (FCT, https://www.fct.pt/) (SFRH/BPD/96176/2013 CC, and IF/00735/2014 to AC). YY, MS and AB were supported by a Deutsche Forschungsgemeinschaft (DFG, https://www.dfg. de/) CRC/TRR 124, project A3. MRF acknowledges support by La Science Pour la Santé ATIP Avenir (https://www.inserm.fr/connaitre-inserm/laureats-atip-avenir) and Fondation pour la Recherche Médicale (FRM, https://www.frm.org/) Equipe grant. The funders had no role in study design, data collection and analysis, decision to publish, or preparation of the manuscript.

**Competing interests:** I have read the journal's policy and the authors of this manuscript have the following competing interests. In the past 5 years SG has received research funds from Pfizer and has been a council member of the International Society of Human and Animal Mycology (ISHAM). All other authors declare no conflicts of interest.

**Abbreviations:** AM, alveolar macrophage; BSF, Biological Services Facility; CBS, cystathionine β-synthase; CMV, cytomegalovirus; CTH, cystathionine γ-lyase; DMEM, Dulbecco's Modified Eagle Medium; ELISA, enzyme-linked immunosorbent assay; EORTC, European Organization for Research and Treatment of Cancer; FBS, fetal bovine serum; GAPDH, glyceraldehyde 3-phosphate dehydrogenase; GO, Gene Ontology; GVHD, graft-versus-host-disease; $H_2S$, hydrogen sulfide; IA, invasive aspergillosis; IC, immunocompetent; IPA, invasive pulmonary aspergillosis; IPO, Instituto Português de Oncologia; LC–MS/MS, liquid chromatography–tandem mass spectrometry; MDM, monocyte-

## Introduction

Posttranslational modifications (PTMs) constitute a rapidly acting response mechanism that permits fast adaptation to short-lasting and varying stresses. Therefore, appropriately timed and executed PTMs are likely crucial for the survival of pathogens inside their hosts [1] as well as for optimal host responses [2]. Hydrogen sulfide ($H_2S$) is a gaseous signaling molecule or gasotransmitter, produced in mammalian tissues by at least 3 enzymes—cystathionine β-synthase (CBS), cystathionine γ-lyase (CTH), and 3-mercaptopyruvate sulfurtransferase (MST) [3,4]. It has been postulated that $H_2S$ exerts its signaling via protein persulfidation [5], a PTM that consists of the conversion of a thiol (−SH) into a persulfide (−SSH) group in cysteine residues of target proteins [6]. The exact mechanism by which $H_2S$ becomes activated to modify specifically targeted cysteine residues remains unclear [7]. Persulfidation can increase or decrease the function or activity of a given protein, which translates into a prominent regulatory role for various physiological functions [8], including inflammation and counteracting endoplasmic reticulum stress [5]. Furthermore, the number of proteins discovered to undergo persulfidation is steadily increasing [8]. Nevertheless, despite evidence of its importance, little is known about the role and relevance of protein persulfidation in immune responses to pathogen challenge.

In contrast to extensive research undertaken on $H_2S$ signaling and persulfidation in mammalian cells, insights about their relevance in microbes are limited. $H_2S$ production has been shown to be important for antibiotic susceptibility of several bacteria [9] and their defense against the host immune response [10] and inflammation [11]. To date, the relevance of persulfidation, specifically, has only been studied in *Staphylococcus aureus*, where it was linked to resistance against antibiotics, cellular redox stress, and the global regulation of the production of virulence factors [12].

Herein, we address the relevance of persulfidation for the adaptation of the human pathogenic fungus *Aspergillus fumigatus* to its mammalian host and for host defense against pathogen challenge. *A. fumigatus* produces millions of airborne spores that, due to their small size, can penetrate the human respiratory tract. Inhalation of *A. fumigatus* spores rarely has adverse effects in immunocompetent (IC) individuals, since the spores are efficiently eliminated by host innate immunity. However, immune disorders may lead to a spectrum of diseases collectively named aspergilloses [13,14]. In Europe alone, the number of clinical conditions caused by *A. fumigatus* exceeds 2 millions cases per year, including an estimated 50,000 cases of life-threatening invasive aspergillosis (IA) [15]. The latest estimates calculate a global incidence for IA higher than 300,000 (suspected to be an underestimate) and a mortality rate ranging from 30% to 80% [16].

We reveal that disruption of the CTH encoding gene in either *A. fumigatus* or human alveolar epithelial cells diminishes their protein persulfidation levels. Reduced protein persulfidation in *A. fumigatus* is correlated with decreased virulence, as we show that this PTM modulates peroxiredoxin and alcohol dehydrogenase activities, which are known to be relevant for fungal pathogenicity. Furthermore, we demonstrate that normal levels of host persulfidation are required for maximum antifungal potency of lung resident alveolar macrophages (AMs) and epithelial cells. This correlates with the observed higher incidence of invasive pulmonary aspergillosis (IPA) in hematopoietic stem cell transplant recipients carrying a single nucleotide polymorphism (SNP) in the gene coding for CTH. Finally, we show that the extent of host protein persulfidation, which directly correlates with its capacity to defend against *A. fumigatus* infection, determines the level of persulfidation that *A. fumigatus* requires counteract the action of the host during the course of infection.

derived macrophage; MM, minimal medium; MSG, Mycology Study Group; MST, 3-mercaptopyruvate sulfurtransferase; NBF-Cl, 4-chloro-7-nitrobenzofurazan; PBMC, peripheral blood mononuclear cell; PCR, polymerase chain reaction; PDA, potato dextrose agar; PI, propidium iodide; PMA, phorbol 12-myristate 13-acetate; PTM, posttranslational modification; PVDF, polyvinylidene difluoride; qPCR, quantitative real-time PCR; RSLC, Rapid Separation LC; RT, room temperature; SD, standard deviation; SECVS, Ethics Subcommittee for Life and Health Sciences; SNP, single nucleotide polymorphism.

## Results

### Persulfidation cannot be abrogated in *Aspergillus fumigatus*

*A. fumigatus* orthologue proteins CTH (*mecB*, AFUA_8G04340), CBS (*mecA*, AFUA_2G07620), and MST (*mstA*, AFUA_8G01800) were identified by BLASTp [17] of the *A. fumigatus* Af293 proteome (taxid:330879) using the well-characterized human proteins (UniProtKB IDs P32929, P35520, and P25325) as queries. MecB and MecA are highly similar to their human counterparts (MecB 53% identity, 69% similarity; MecA 54% identity, 69% similarity) indicating a conserved activity, while MstA has a lower similarity rate (37% identity, 51% similarity), suggesting the potential of having an analogous function. Indeed, all 3 fungal proteins retrieved the same functional classification as their human counterparts in InterProScan [18], strongly indicating that they are functional orthologues.

To gain insight into the intrinsic control of persulfidation in *A. fumigatus*, we constructed deletion strains for the 3 identified genes by homologous gene replacement in the wild-type ATCC46645 strain, employing a self-excising recyclable marker [19]. Persulfidation levels were monitored by both in-gel detection (Fig 1A) and fluorescence microscopy (Fig 1B), using the dimedone switch method [20]. Relative to the wild type, quantification of persulfidation levels in whole hyphal protein extracts revealed significantly decreased persulfidation in *ΔmecA* (33% reduction $P = 0.005$) and *ΔmecB* (42% reduction, $P = 0.001$) deletion mutants (Fig 1A). Reconstitution of the *mecB* gene in its native locus reverted the persulfidation levels to those of the wild type ($P = 0.41$, S1A Fig). Microscopy-mediated quantification of persulfidation levels in hyphae grown in minimal medium (MM) and in Dulbecco's Modified Eagle Medium (DMEM) revealed a reduction, relative to wild type, in the *ΔmecA* and *ΔmecB* mutants, which was statistically significant for the latter ($P = 0.001$ in MM and $P = 0.046$ in DMEM) (Fig 1B). Aiming to reduce the levels of persulfidation further, we attempted to construct a double *ΔmecA ΔmecB* mutant by targeting each gene for deletion in the corresponding single mutant strain but repeatedly failed. We therefore tested if loss of function in both genes could have a synthetic lethal phenotype by using the heterokaryon rescue technique, a method designed for the identification of essential genes in *Aspergillus* species [21]. We indeed observed that conidia from primary transformants could be propagated in nonselective media but not in selective medium, indicating that the double transformant nuclei can only be maintained in heterokaryosis and therefore that loss of function of both gene products has a synthetic lethal outcome. Aiming to validate this hypothesis, we constructed strains in each mutant background placing the other gene under the control of the TetOFF promoter (i.e., *mecBΔmecA_tetOFF* and *mecAΔmecB_tetOFF*), which we have recently and successfully used to investigate the function of an essential gene [22]. However, the basal expression of the genes under the TetOFF control was so high that addition of doxycycline could not down-regulate their transcription to levels significantly below of the native promoter (S1B Fig). Accordingly, the strains grew normally in restrictive conditions (S1C Fig), and their persulfidation levels were similar to that of the single mutants (S1D Fig). Therefore, even if further investigations are needed to formally confirm it, the current evidence suggests that the genes *mecA* and *mecB* are synthetically lethal.

The mutants showed no phenotype during normal growth in submerged or solid conditions. We evaluated the sensitivity of all 3 single mutants to a variety of common stressors (S2 Fig) and found that none of the mutants was sensitive to high temperature (48°C), hypoxia (1% $O_2$), osmotic stress (NaCl or KCl), or cell wall disturbing agents (Congo Red, Calcofluor White, or Caffeine) (S2A Fig). The *ΔmecA* mutant was slightly sensitive than wild type to the cell wall stressor SDS (S2A Fig). Remarkably, all mutants were more sensitive to $H_2O_2$ than the wild type (S2B Fig) and Fludioxonil, an antifungal which action is enhanced when glutathione

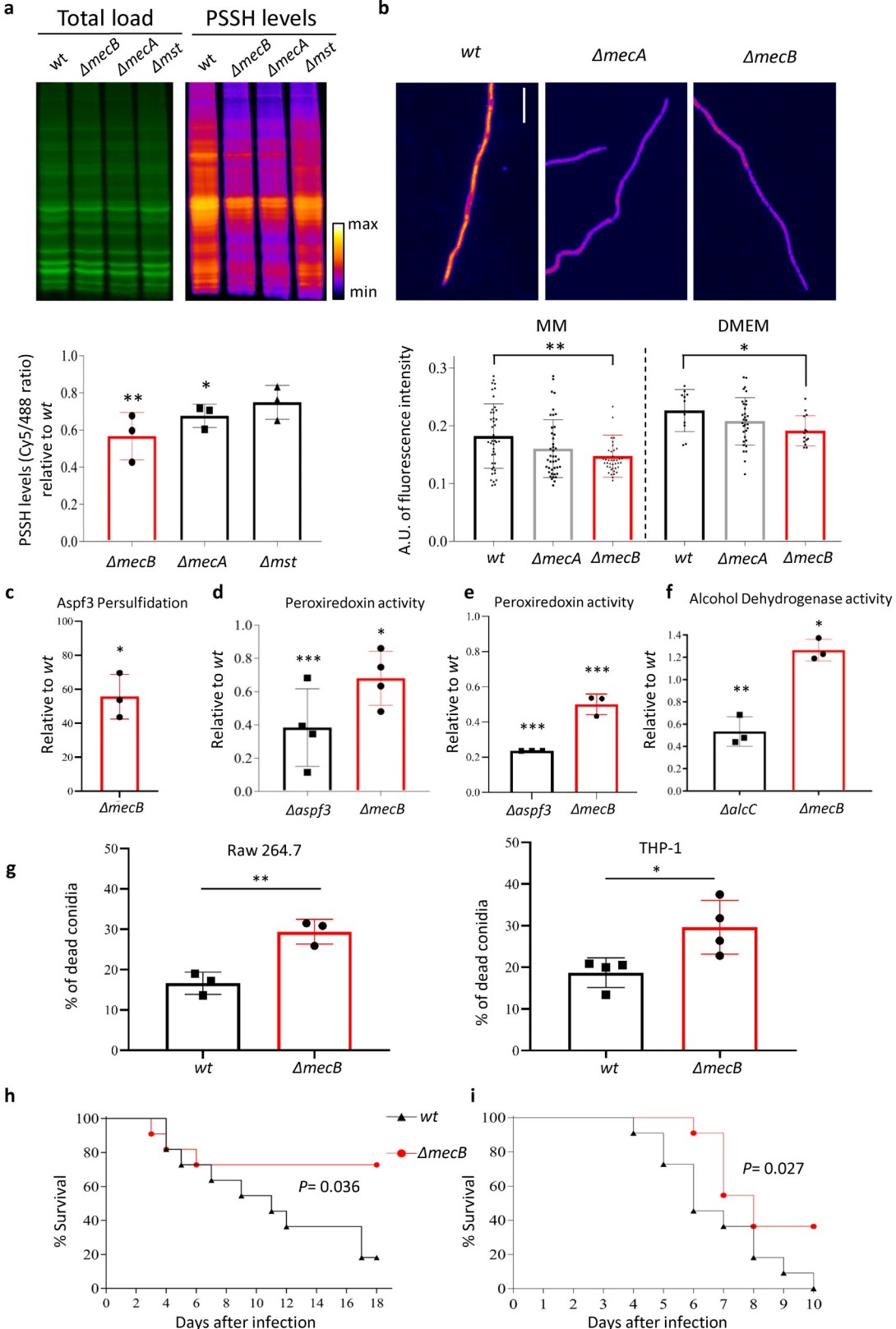

**Fig 1. Persulfidation modulates the activity of *A. fumigatus* virulence–associated proteins and is important for its pathogenic potential. (a)** Representative images of in-gel detection of persulfidation levels in whole protein extracts of *A.*

*fumigatus* wt and mutants *ΔmecB* (CTH), *ΔmecA* (CBS), and *ΔmstA* (MST). NBF-Cl labels persulfides, thiols, sulfenic acids, and amino groups; reaction with amino groups produces the green signal, therefore it reflects the whole protein context and is used to normalize the persulfidation levels. The red signal is produced by the dimedone-Cy5 labeled probe, which selectively switches NBF adduct on persulfide groups [20]. Quantification of persulfidation levels, measured as the ratio of red signal normalized to the green signal, revealed a significant decrease in persulfidation level of *ΔmecB* relative to wild type ($n = 3$). **(b)** Representative images (scale bar = 15 μm) and quantification of persulfidation levels measured by microscopy showed a significant decrease in *ΔmecB*. The intensity of fluorescence is represented as 32-bit arbitrary units ($n = 3$, >10 photos = hyphae per sample). **(c)** The ratio of persulfidated Aspf3 (normalized to total Aspf3 levels, S2B Fig) was significantly decreased in the *ΔmecB* mutant relative to wild type, as analyzed using a 1-sample *t* test ($n = 3$). **(d)** In a peroxiredoxin enzymatic assay, *ΔmecB* and *Δaspf3* showed a significant ($P = 0.008$) decrease of activity compared to wild type ($n = 4$, with 3 technical replicates). Data were analyzed using a 1-way ANOVA with Dunnett multiple comparisons. **(e)** In a thioredoxin-dependent assay, *ΔmecB* and *Δaspf3* also showed a significant ($P < 0.0001$) decrease of activity compared to wild type ($n = 3$, with 3 technical replicates). Data were analyzed using a 1-way ANOVA with Dunnett multiple comparisons. **(f)** In an alcohol dehydrogenase enzymatic assay, *ΔalcC* had a significantly decreased and *ΔmecB* increased activity in comparison with wild type ($n = 3$, with 3 technical replicates). All data are depicted as mean ± SD and were analyzed using a 1-way ANOVA with Dunnett multiple comparisons. **(g)** *ΔmecB* was more sensitive to killing by murine Raw.264.7 macrophages (unpaired 2-tailed *t* test, $n = 3$) and human THP-1 macrophages (unpaired 2-tailed *t* test, $n = 4$) (both assays were run with 3 technical replicates). **(h)** The *ΔmecB* strain showed a significant reduction in virulence in a corticosteroid model of IPA ($P = 0.036$ log-rank test) and **(i)** in a leukopenic murine model of IPA ($P = 0.027$ log-rank test) ($n = 11$ animals per group). All numerical values that underlie the data displayed in this figure can be found in S1 Data. CBS, cystathionine β-synthase; CTH, cystathionine γ-lyase; DMEM, Dulbecco's Modified Eagle Medium; IPA, invasive pulmonary aspergillosis; MM, minimal medium; MST, 3-mercaptopyruvate sulfurtransferase; NBF-Cl, 4-chloro-7-nitrobenzofurazan; PSSH, Persulfidation; SD, standard deviation; wt, wild-type.

homeostasis is compromised [23] (S2B Fig). In addition, *ΔmecB* was more sensitive to menadione and slightly to the thiol-oxidizing drug diamide (S2B Fig). Reconstitution of *mecB* in its natural locus (*mecB+*) restored wild-type resistance to all oxidative stressors (S2C Fig). Persulfidation is known to be important for cellular redox processes (for a review, see [24]) due to its role in preventing cysteine hyperoxidation [20]. Therefore, it is not surprising that reduced levels of persulfidation cause sensitivity to oxidative stressors in *A. fumigatus*, as has been described in other organisms [20]. The *ΔmecB* mutant displayed no phenotype when grown on solid medium containing methionine or cysteine as sole S-sources (S2D Fig), suggesting that disturbance of the trans-sulfuration pathway is not impactful on the homeostasis of sulfur metabolism. Finally, the *ΔmecB* mutant showed the same susceptibility profile as the wild type to the antifungals amphotericin B, voriconazole, and anidulafungin, respectively representing polyenes, azoles, and echinocandins (S1 Table), suggesting that persulfidation is not important for *A. fumigatus* antifungal resistance mechanisms.

Therefore, to investigate if persulfidation dependent responses are relevant for pathogen adaptation to host conditions, we selected *ΔmecB*, the *A. fumigatus* mutant with the most reduction in low-level persulfidation for subsequent analyses. It is important to note that all phenotypes displayed by the *ΔmecB* mutant are due to an approximately 45% reduction of persulfidation, not to the absolute absence of this PTM.

## Persulfidation affects enzymatic activities known to be relevant for *A. fumigatus* pathogenicity

The use of 3 independent methodologies, the dimedone switch assay [20], the biotin thiol assay [25], and the improved switch tag technique [26], confirmed that many proteins are persulfidated in *A. fumigatus* (Fig 1A, S3A Fig). Seeking to gain insight of the persulfidated proteome in *A. fumigatus*, we enriched the persulfidated fraction using the most selective method, the dimedone switch assay [20], and identified proteins by mass spectrometry (S2 Table). We run a pathway enrichment analysis with the identified proteins using the YeastEnrichr platform [27,28]. This analysis showed an enrichment of the process translation in the Gene Ontology (GO) classifications biological process and molecular function (S3B Fig). This suggests that persulfidation may influence translation and therefore have an impact on the

proteome content. Among the persulfidated proteins, many previously described to be important for *A. fumigatus* pathogenic potential can be found. We concentrated on the effect of persulfidation on peroxiredoxins, as two of the detected peroxiredoxins Aspf3 (UniProt: A0A0J5PFY6; Gene ID: AFUA_6G02280) and PrxA (UniProt: A0A0J5PIP5; Gene ID: AFUA_4G08580) are strictly required for *A. fumigatus* pathogenicity [29,30]. To check if the level of persulfidated Aspf3 is diminished in the *ΔmecB* isolate compared to the wild-type progenitor, we enriched the persulfidation fraction of full protein lysates and specifically detected Aspf3 by western blot (S3C Fig). The ratio of Aspf3 in the enriched fraction normalized to the full extract confirmed that its persulfidation is indeed reduced by approximately 45% in the *ΔmecB* mutant ($P = 0.028$; Fig 1C). In order to investigate whether reduced persulfidation affect the activity of peroxiredoxins, we measured $H_2O_2$ detoxifying activity using 2 enzymatic assay, one based on total mycelia extracellular degradation of tert-butyl hydroperoxide [31] (S4A Fig) and a thioredoxin-dependent assay with total protein lysates [32] (S4B Fig). In both cases, *A. fumigatus ΔmecB* showed a significant decrease in $H_2O_2$ detoxifying activity relative to the wild type, of approximately 35% using the extracellular assay ($P = 0.008$, Fig 1D) and of approximately 50% using the peroxiredoxin thioredoxin-dependent assay ($P < 0.0001$, Fig 1E), demonstrating that reduced persulfidation levels in the fungus impacted $H_2O_2$ detoxifying activity, including peroxiredoxin activity. An overall reduced peroxiredoxin activity in low persulfidation agrees with a previous study that reported how persulfidation protects peroxiredoxins from irreversible oxidation, therefore maintaining their function [33]. To better understand the consequences for this decrease in $H_2O_2$ detoxifying activity, we measured the ratio of monomeric to dimeric Aspf3 upon $H_2O_2$ treatment and found that it was higher in *ΔmecB* compared to the wild type (S4C Fig), indicating a higher oxidation level of the protein [34]. Indeed, the total levels of protein oxidation (sulfenylation, PSOH) and hyperoxidation (sulfinylation, $PSO_2H$) upon $H_2O_2$ treatment were higher in *ΔmecB* compared to the wild type (S4D and S4E Fig), indicating a higher oxidation status in the cells. We then examined if reduced levels of persulfidation could cause higher levels of Aspf3 hyperoxidation (sulfinylation, $PSO_2H$), but did not find any evidence that Aspf3 suffered this type of inactivation (S4F Fig), which is not surprising as the human orthologue Prx5 is known not to undergo hyperoxidation [35]. Various alcohol dehydrogenases were also identified as persulfidated proteins (S2 Table), and one of them, AlcC (UniProt: A0A0J5PVH3; Gene ID: AFUA_5G06240), has been implicated in *A. fumigatus* virulence [36]. Therefore, we also measured alcohol dehydrogenase activity from crude protein extracts and found that the activity was significantly increased by approximately 25% ($P = 0.025$) in the low persulfidation *ΔmecB* mutant, compared to the wild-type strain, while the activity of the *ΔalcC* strain was strongly diminished, as expected (Fig 1F, S5A Fig).

## Correct persulfidation levels are relevant for *A. fumigatus* pathogenic potential

Given the importance of Aspf3 for oxidative stress resistance during infection [29] and the observed higher susceptibility of *ΔmecB* to oxidative stressors (S2 Fig), we speculated that the lower peroxiredoxin activity in the *ΔmecB* mutant would translate into a higher susceptibility to killing by immune effector cells. Therefore, we performed spore killing assays using murine (Raw.264.7) and human (THP-1) macrophage cell lines. Both cell lines killed *ΔmecB* conidia to a significantly higher level than wild-type conidia (29.4% versus 16.6% for Raw.264.7, $P = 0.057$ and 29.6% versus 18.7% for THP-1, $P = 0.025$) (Fig 1G). The reconstituted *mecB+* strain was again killed by Raw.264.7 macrophages at the same ratio as the wild type ($P = 0.92$, S5B Fig), indicating that the effect is specific to *mecB* deletion. Conidia encounter a challenging nutritional environment inside phagocytes, and thus it could be that the mutant's higher

susceptibility to killing was related to the metabolic role of *mecB*; however, given that *ΔmecB* showed no phenotype on plates containing single S-sources (S2D Fig), we believe this is unlikely. Alternatively, the shown altered activity of fungal proteins known to be relevant for virulence in the *ΔmecB* mutant can be expected to cause higher susceptibility to killing by effector immune cells. We then investigated the *ΔmecB* pathogenic potential in 2 well-established murine models of IPA [37] (Fig 1H and 1I). The *ΔmecB* mutant showed a significant reduction in virulence compared to the wild-type strain in both the corticosteroid ($P = 0.036$, Fig 1G) and the leukopenic ($P = 0.027$, Fig 1H) models of immunosuppression. In addition, fungal burden at 3 days postinfection was significantly lower in leukopenic mice infected with *ΔmecB* than in mice infected the wild-type strain ($P = 0.03$, S5B Fig).

In summary, these results demonstrate that a moderate reduction of approximately 45% in *A. fumigatus* persulfidation correlates with a significant decrease in its virulence. This reduction is likely due to pleiotropic effects caused by the imbalanced activity of many proteins, including, but not limited to, peroxiredoxins.

## A SNP in the human cystathionine γ-lyase encoding gene is associated with higher incidence of invasive pulmonary aspergillosis in hematopoietic stem cell transplant recipients

In humans, improper function of the $H_2S$ producing enzyme CTH has been implicated in various hyperinflammatory conditions [38,39]. Accordingly, we speculated that optimal CTH activity and therefore adequately modulated persulfidation levels (CTH is the main responsible for persulfidation in lung tissues [40]) must be required to mount a proper response against invading respiratory pathogens. To investigate our hypothesis, we searched for reported SNPs in the CTH open reading frame that could potentially affect protein function. Indeed, there is one relatively common (MAF T = 0.21 to 0.49 according to DbSNP), non-synonymous SNP previously described in the CTH encoding gene (NG_00804): SNP S403I (G>T; rs1021737). To test if this SNP affects protein function, which has not been investigated before, we measured the relative activity of purified recombinant CTH proteins expressed in *Escherichia coli* with and without the SNP. We observed that $H_2S$ production was reduced by 18% in the recombinant protein carrying the S403I variant ($P < 000.1$, Fig 2A), implying that the SNP causes a reduction in protein activity. To check if this decreased activity results in reduced persulfidation levels in cells, we measured persulfidation levels in monocyte-derived macrophages (MDMs) from healthy donors with and without the SNP. We found that the basal persulfidation levels were slightly elevated in cells that carry TT genotype compared to the reference GG genotype (Fig 2B), expectedly due to a compensatory mechanism. Interestingly, upon challenging MDMs from the GG genotype with *A. fumigatus* conidia, we observed a strong increase in their persulfidation levels, whereas the response in MDMs with the TT genotype was completely blunted (Fig 2B). This lack of response might be directly due to the lower activity of the CTH with SNP or to a defective signaling mechanism in which CTH itself could be implicated. Additional experiments with more donors will be required to validate this result and understand the underlying mechanism. We then tested if MDMs carrying the TT genotype have an dysregulated cytokine production, as the activity of proteins that regulate the immune response and cytokine production are modulated by persulfidation, like NFκB [41] and tristetraprolin [42]. Indeed, we found that TT MDMs produced a significantly lower amount of pro-inflammatory cytokines when challenged with *A. fumigatus* conidia than cells with the GG or heterozygous genotypes (Fig 2C).

Given that presence of the SNP showed functional consequences, we decided to investigate the relationship between the genetic variability of *CTH* and the incidence of IPA after

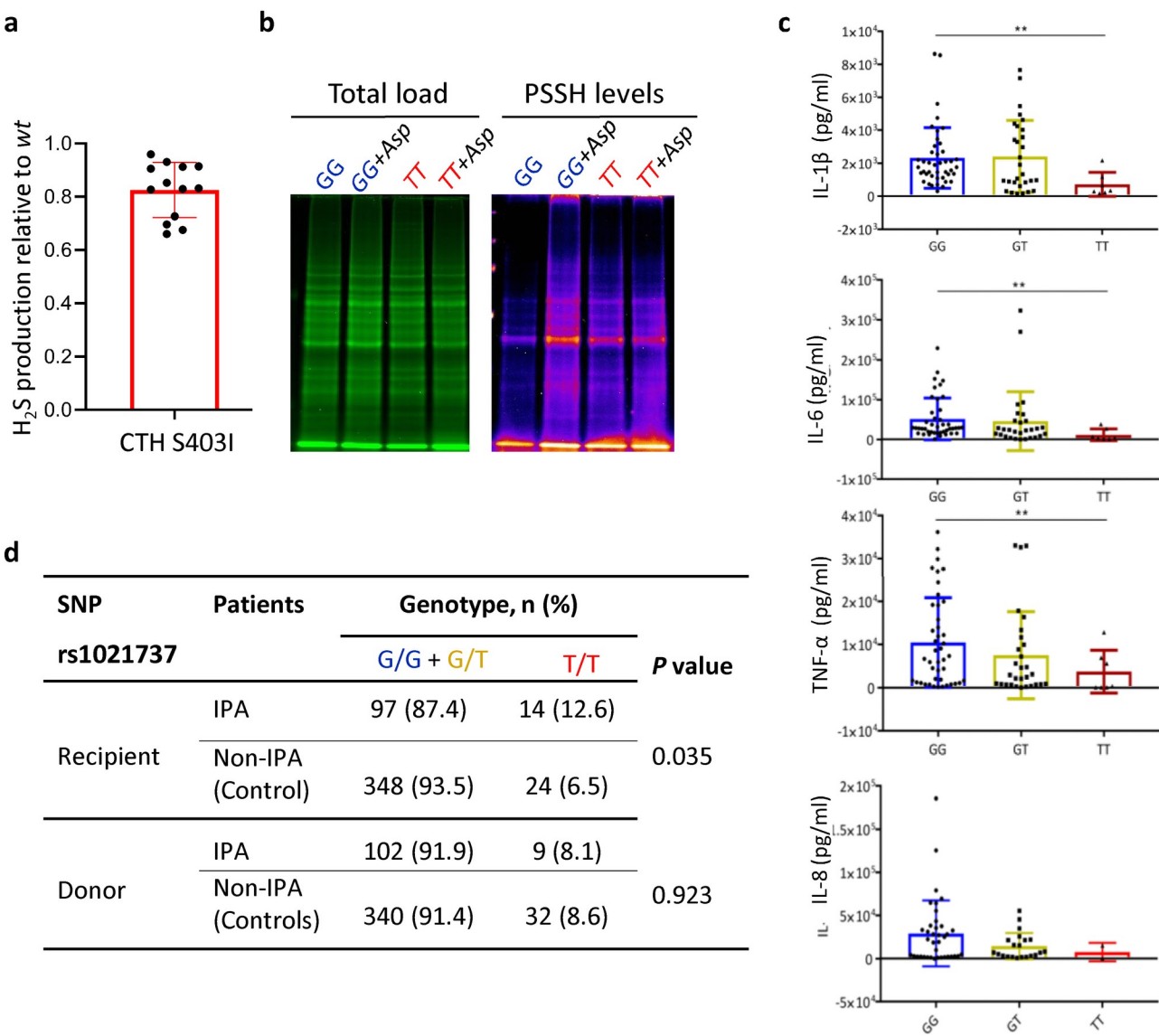

**Fig 2. A SNP in the human CTH is associated with higher incidence of IPA in hematopoietic stem cell transplant recipients. (a)** Recombinant CTH enzyme carrying the SNP S403I (rs1021737) had a significantly reduced enzymatic activity compared to wt enzyme, measured as relative production of $H_2S$ ($P = 0.0001$, 1-sample $t$ test) ($n = 3$ independent experiments with 4 replicates). **(b)** The persulfidation level of human MDMs with the SNP (TT) was slightly elevated compared to wt genotype (GG). Upon challenge with *A. fumigatus* conidia (Asp), the persulfidation level of GG MDMs was strongly increased, while the level in TT MDMs remained unchanged ($n = 1$). **(c)** Upon challenge with *A. fumigatus*, production of proinflammatory cytokines was lower in TT MDMs compared with wt (GG) and heterozygous (GT) genotypes (each point in the graph represents cytokine production by MDMs from 1 healthy donor). **(d)** Presence of the SNP in homozygosis in the recipient was more common in patients that developed IPA (12.6%) than in patients that did not (controls, 6.5%) (Fisher extract $t$ test $P = 0.035$). This translated into an overall increase in the incidence of IPA of approximately 15% in the transplant recipients carrying the SNP in homozygosis (21.7% in GG+GT versus 36.8% in TT). No effect of the presence of the SNP in the donor compartment was detected. All numerical values that underlie the data displayed in this figure can be found in S2 Data. CTH, cystathionine γ-lyase; $H_2S$, hydrogen sulfide; IPA, invasive pulmonary aspergillosis; MDM, monocyte-derived macrophage; SNP, single nucleotide polymorphism; wt, wild-type.

hematopoietic stem cell transplantation. We did not detect any effect of the presence of the SNP in the donor compartment on the incidence of IPA in transplant recipients. Intriguingly, we observed that in transplant recipients, TT genotype was more frequently found in IPA patients than in controls (12.6% in IPA versus 6.5% in controls, $P = 0.035$, Fig 2D), and

consequently, the overall incidence of IPA was higher in carriers of the TT genotype than in other genotypes (36.8% for TT versus 21.7% GG+GT). Analysis of the baseline characteristics of the transplant recipients showed that the variables associated with a higher incidence of IPA were the type of transplant, acute graft-versus-host-disease (GVHD), and antifungal prophylaxis (Table 1). Interestingly, the only baseline characteristic that associated with the SNP in CTH was a positive serostatus for CMV (cytomegalovirus) (Table 2). This agrees with previous studies that reported a relevant role of CTH and $H_2S$ in respiratory viral infections in mice [43] and airway epithelial cells [44,45]. These results suggest that there is an association of the presence of the SNP in transplant recipients with the incidence of IPA.

## Low persulfidation levels correlate with a decrease in the antifungal potency of alveolar macrophages and epithelial cells

The fact that presence of the CTH SNP S403I in the recipient is associated with an increase in IPA incidence but its presence in the donor has no effect on the incidence of IPA suggests that correct persulfidation levels are more important for mounting a proper antifungal response in lung non-hematopoietic cells (e.g., epithelial cells) and lung resident immune cells that are relatively resistant to conditioning treatments (and therefore are partially recipient derived for long periods of time, e.g., AMs [46–49]). To investigate this hypothesis, we made use of the CTH knock-out C57BL/6$^{CTH-/-}$ [50] mouse line, which was previously demonstrated to have decreased levels of persulfidation in the lung tissue [26]. We isolated bone marrow neutrophils (representative of donor-derived cells in a transplant recipient) and AMs (representative of lung resident recipient cells) from C57BL/6 and C57BL/6$^{CTH-/-}$ and challenged them with *A. fumigatus* wild-type and *ΔmecB* conidia ex vivo. Both immune effector cells killed *ΔmecB* spores to a significantly higher degree than wild-type conidia (Fig 3A), further supporting the relevance of persulfidation for *A. fumigatus* capacity to counteract host attack. Interestingly, C57BL/6$^{CTH-/-}$ AMs, but not neutrophils, showed a defect in conidial killing (approximately 11% for wild type *P* = 0.0003 and approximately 5% for *ΔmecB P* = 0.078), which supports the notion that correct persulfidation is required for the antifungal action of lung resident cells but not of donor-derived type cells. To further investigate this hypothesis, we next measured cytokines in lung homogenates of IC and leukopenic C57BL/6 and C57BL/6$^{CTH-/-}$ mice, both in steady state conditions and 24 hours after *A. fumigatus* challenge (S6 Fig). The levels of pro-inflammatory cytokines in C57BL/6 IC infected mice were low (S6 Fig). This was expected as 24 hours after challenge, IC mice should have cleared infection and resolved inflammation. Notably, the levels of pro-inflammatory cytokines were slightly higher in C57BL/6$^{CTH-/-}$, which could indicate that elimination of the fungus is less efficient in this strain and thus the production of cytokines is sustained for a longer period. Interestingly, leukopenic C57BL/6$^{CTH-/-}$ mice challenged with *A. fumigatus* had significantly lower levels of IL-1α and TNF-α than leukopenic C57BL/6 mice (Fig 3B, S6 Fig), which reflects a defective response to fungal challenge in this condition. Leukopenic mice are depleted of proliferative hematopoietic cells (reflecting the donor compartment of a transplant) but not of lung resident cells as AMs and epithelial cells (reflecting the recipient compartment). Therefore, the reduced cytokine production in leukopenic C57BL/6$^{CTH-/-}$ compared with leukopenic C57BL/6 upon infection can be attributed specifically to a defect in the antifungal response of lung resident cells, which is in agreement with the increase in the incidence of IPA when the SNP in CTH is present in the recipient compartment. To further validate this conclusion, we attempted survival experiments, but we found that delivery of *A. fumigatus* conidia both intranasally and intratracheally in C57BL/6 wild type and knock-out caused an extremely high number of cases with balance problems (up to 25%, when the common incidence is ≤1%). This is likely caused by high

**Table 1. Baseline characteristics of transplant recipients enrolled in the study.**

| Variables | IPA (*n* = 111) | No IPA (*n* = 372) | *P* value |
|---|---|---|---|
| **Age at transplantation, no (%)** | | | |
| ≤20 years | 16 (14.4) | 81 (21.8) | 0.150 |
| 21–40 years | 30 (27.0) | 108 (29.0) | |
| >40 years | 65 (58.6) | 183 (49.2) | |
| **Gender, no (%)** | | | |
| Female | 48 (43.2) | 158 (42.5) | 0.870 |
| Male | 63 (56.8) | 214 (57.5) | |
| **Underlying disease, no. (%)** | | | |
| Acute leukemia | 61 (55.0) | 197 (53.0) | 0.223 |
| Chronic lymphoproliferative diseases | 16 (14.4) | 67 (18.0) | |
| Myelodysplastic/myeloproliferative diseases | 17 (15.3) | 34 (9.1) | |
| Chronic myeloproliferative diseases | 8 (7.2) | 21 (5.6) | |
| Aplastic anemia | 6 (5.4) | 29 (7.8) | |
| Other | 3 (2.7) | 24 (6.5) | |
| **Transplantation type, no. (%)** | | | |
| Matched, related | 36 (32.4) | 175 (47.0) | 0.009 |
| Matched, unrelated | 40 (36.0) | 91 (24.5) | |
| Mismatched, related | 0 (0.0) | 8 (2.2) | |
| Mismatched, unrelated | 35 (31.5) | 98 (26.3) | |
| **Graft source, no. (%)** | | | |
| Peripheral blood | 91 (82.0) | 306 (82.3) | 0.645 |
| Bone marrow | 19 (17.1) | 57 (15.3) | |
| Cord blood | 1 (0.9) | 9 (2.4) | |
| **Disease stage, no. (%)** | | | |
| First complete remission | 59 (53.2) | 204 (54.8) | 0.940 |
| Second or subsequent remission, or relapse | 19 (17.1) | 63 (17.0) | |
| Active disease | 33 (29.7) | 105 (28.2) | |
| **Conditioning regimen, no (%)** | | | |
| RIC | 79 (71.2) | 250 (67.2) | 0.452 |
| Myeloablative | 32 (28.8) | 122 (32.8) | |
| **CMV serostatus of donor and recipient, no. (%)** | | | |
| D−/R+ or D+/R+ | 94 (84.7) | 331 (89.0) | 0.214 |
| D−/R− or D+/R− | 17 (15.3) | 41 (11.0) | |
| **Duration of neutropenia, mean days (range)**[†] | 13.2 (8–39) | 14.0 (5–35) | 0.504 |
| **Acute GVHD, no. (%)** | | | |
| No GVHD or grades I and II | 77 (69.4) | 325 (87.4) | <0.001 |
| Grades III and IV | 34 (30.6) | 47 (12.6) | |
| **Antifungal prophylaxis, no. (%)**[‡] | | | |
| Fluconazole | 55 (49.6) | 149 (40.1) | 0.036 |
| Posaconazole | 31 (27.9) | 120 (32.3) | |
| Other | 9 (8.1) | 15 (4.0) | |
| None or unknown | 16 (14.4) | 88 (23.7) | |

Chronic lymphoproliferative diseases included cases of chronic lymphocytic leukemia, multiple myeloma, and B cell and T-cell lymphomas. Chronic myeloproliferative diseases included cases of chronic myelogenous leukemia and primary myelofibrosis. Other diseases included cases of idiopathic medullar aplasia, lymphohistiocytosis, hemoglobinopathies, and paroxysmal nocturnal hemoglobinuria.

[†]Neutropenia was defined as ≤0.5 × 10$^9$ cells/L.

[‡]Other antifungals used in prophylaxis included voriconazole, liposomal amphotericin B, itraconazole, and caspofungin.

*P* values were calculated by Fisher exact probability *t* test or Student *t* test for continuous variables.

CMV, cytomegalovirus; D, donor; GVHD, graft-versus-host-disease; IPA, invasive pulmonary aspergillosis; R, recipient; RIC, reduced intensity conditioning.

deposition of spores in the sinus of C57BL/6 mice, rather than in the lung parenchyma, probably due to anatomical particularities of their airway structure [51], which are known to influence particle deposition [52]. Therefore, in C57BL/6 the fungus seems to invade the sinus and surrounding tissues, in which persulfidation levels can be influenced by other enzymes (as CBS) [5] and, consequently the relevance of CTH-dependent persulfidation levels in the lungs for mouse survival cannot be investigated using these mice.

Aiming to investigate the contribution of non-hematopoietic cells in more detail, we disrupted the CTH encoding gene in the human alveolar epithelial cell line A549 using CRISPR/Cas-9. Western blot phenotype confirmed a molecularly-demonstrated knock-out genotype of the resulting $CTH^{-/-}$ cell line (S7A Fig), which showed a significant reduction in persulfidation (S7B Fig). As in *A. fumigatus*, persulfidation was not completely abrogated in this cell line, hence the effects described below are correlated with the significant reduction of approximately 55% in the persulfidation levels. We challenged A549 and $CTH^{-/-}$ cell lines with *A. fumigatus* wild-type and *ΔmecB* strains and measured epithelial cell detachment to evaluate the degree of host damage incurred by the pathogen (Fig 3C). Interestingly, we observed that the *A. fumigatus ΔmecB* strain induced slightly less detachment than the wild type in A549 cell layer (9.8% lesser, $P = 0.097$ not significant) and significantly less in $CTH^{-/-}$ cell layer (21.1% lesser, $P = 0.0013$). In addition, the $CTH^{-/-}$ cell monolayer suffered significantly higher detachment than the A549 monolayer during incubation with wild-type conidia (14.1% more, $P = 0.017$), but not *ΔmecB* spores. This experiment suggests that persulfidation is relevant for both the fungal potential to cause damage as well as the host capacity to withstand assault. To investigate the killing capability of the epithelial cells, we calculated the percentage of dead conidia after 6 hours of co-incubation (Fig 3D) and observed that (1) the *A. fumigatus ΔmecB* conidia were killed to a higher extent than wild type (33.9% versus 20.7% conidia killed by A549 $P < 0.0001$ and 27.2% versus 16.7% by $CTH^{-/-}$ $P = 0.0006$); and (2) the $CTH^{-/-}$ cells were less efficient in killing fungal conidia compared to the progenitor A549 cells (16.7% versus 20.7% killed wild-type conidia $P = 0.009$ and 27.2% versus 33.9% killed *ΔmecB* $P < 0.0001$). Therefore, correct persulfidation levels are important for fungal survival and for the capacity of epithelial cells to kill *A. fumigatus*. Surprisingly, we found that $CTH^{-/-}$ cells had internalized significantly more spores than A549 cells 4 hours after challenge (46.5% versus 10.7%, $P = 0.0003$; Fig 3E, S8 Fig), suggesting that uptake is more efficient in low persulfidation cells. Hence, lower levels of persulfidation in the cell likely perturb pathogen killing mechanisms rather than phagocytosis. Finally, IL-8 production in challenged and unchallenged $CTH^{-/-}$ cells was significantly higher than in progenitor A549 cells (Fig 3F), indicating a deregulation of cytokine production in low persulfidation.

Altogether, our observations suggest that deficiency in host CTH, resulting in less persulfidation, negatively affects the capacity of lung resident cells to respond to and kill *A. fumigatus*, rendering hematopoietic stem transplant recipients at higher risk of IPA.

## Host persulfidation determines the level of persulfidation in *A. fumigatus*

Our observations suggest that normal persulfidation in host cells positively correlates with their capacity to kill *A. fumigatus* and orchestrate an adequate antifungal response, and concomitantly, the level of persulfidation in the fungus determines its capacity to survive in the host. Thus, we hypothesized that the level of persulfidation in host cells (killing capacity) may impact the level of persulfidation that the fungus requires in order to counteract host attack. To test this hypothesis, we measured the ratio of persulfidated Aspf3 (by western blotting the total and the persulfidation enriched protein fractions, as described above), a fungal protein serving as a reporter of persulfidation in *A. fumigatus*, in wild-type and *ΔmecB* strains infecting

**Table 2. Association of CTH genotypes with the baseline characteristics of transplant recipients enrolled in the study.**

| Variables | G/G + G/T (*n* = 445) | T/T (*n* = 38) | *P* value |
|---|---|---|---|
| **Age at transplantation, no (%)** | | | |
| ≤20 years | 90 (20.2) | 7 (18.4) | 0.91 |
| 21–40 years | 128 (28.8) | 10 (26.3) | |
| >40 years | 227 (51.0) | 21 (55.3) | |
| **Gender, no (%)** | | | |
| Female | 189 (42.5) | 17 (44.7) | 0.80 |
| Male | 256 (57.5) | 21 (55.3) | |
| **Underlying disease, no. (%)** | | | |
| Acute leukemia | 236 (53.0) | 22 (57.9) | 0.95 |
| Chronic lymphoproliferative diseases | 76 (17.1) | 7 (18.4) | |
| Myelodysplastic/myeloproliferative diseases | 47 (10.6) | 4 (10.5) | |
| Chronic myeloproliferative diseases | 27 (6.1) | 2 (5.3) | |
| Aplastic anemia | 33 (7.4) | 2 (5.3) | |
| Other | 26 (5.8) | 1 (2.6) | |
| **Transplantation type, no. (%)** | | | |
| Matched, related | 191 (42.9) | 20 (52.6) | 0.28 |
| Matched, unrelated | 119 (26.7) | 12 (31.6) | |
| Mismatched, related | 8 (1.8) | 0 (0.0) | |
| Mismatched, unrelated | 127 (28.5) | 6 (15.8) | |
| **Graft source, no. (%)** | | | |
| Peripheral blood | 365 (82.0) | 32 (84.2) | 0.96 |
| Bone marrow | 70 (15.7) | 6 (5.8) | |
| Cord blood | 10 (2.2) | 0 (0.0) | |
| **Disease stage, no. (%)** | | | |
| First complete remission | 240 (53.9) | 23 (60.5) | 0.55 |
| Second or subsequent remission, or relapse | 78 (17.5) | 4 (10.5) | |
| Active disease | 127 (28.5) | 11 (29.0) | |
| **Conditioning regimen, no (%)** | | | |
| RIC | 303 (68.1) | 26 (68.4) | 0.93 |
| Myeloablative | 142 (31.9) | 12 (31.6) | |
| **CMV serostatus of donor and recipient, no. (%)** | | | |
| D−/R+ or D+/R+ | 397 (89.2) | 28 (73.7) | 0.01 |
| D−/R− or D+/R− | 48 (10.8) | 10 (26.3) | |
| **Duration of neutropenia, mean days (range)[†]** | 13.4 (5–39) | 13.8 (9–26) | |
| **Acute GVHD, no. (%)** | | | |
| No GVHD or grades I and II | 368 (82.7) | 34 (89.5) | 0.31 |
| Grades III and IV | 77 (17.3) | 4 (10.5) | |
| **Antifungal prophylaxis, no. (%)[‡]** | | | |
| Fluconazole | 189 (42.5) | 15 (39.5) | 0.88 |
| Posaconazole | 137 (30.8) | 14 (36.8) | |
| Other | 22 (4.9) | 2 (5.3) | |
| None or unknown | 97 (21.8) | 7 (18.4) | |

Chronic lymphoproliferative diseases included cases of chronic lymphocytic leukemia, multiple myeloma, and B cell and T-cell lymphomas. Chronic myeloproliferative diseases included cases of chronic myelogenous leukemia and primary myelofibrosis. Other diseases included cases of idiopathic medullar aplasia, lymphohistiocytosis, hemoglobinopathies, and paroxysmal nocturnal hemoglobinuria.

[†]Neutropenia was defined as ≤0.5 × $10^9$ cells/L.

[‡]Other antifungals used in prophylaxis included voriconazole, liposomal amphotericin B, itraconazole, and caspofungin.

*P* values were calculated by Fisher exact probability *t* test or Student *t* test for continuous variables.

CMV, cytomegalovirus; CTH, cystathionine-γ-lyase; D, donor; GVHD, graft-versus-host-disease; R, recipient; RIC, reduced intensity conditioning.

A549 or $CTH^{-/-}$ cell lines (Fig 4A, S9A Fig). As expected, the level of persulfidated Aspf3 was always higher in the wild-type fungus compared to $\Delta mecB$. Interestingly, the amount of persulfidated Aspf3 was higher when infecting A549 compared to $CTH^{-/-}$, which suggests that a higher persulfidation in host cells triggers more persulfidation in the pathogen. This increase in persulfidation upon contact with the host cells was significant in *A. fumigatus* wild type (increment of 1.62× on A549 relative to $CTH^{-/-}$, $P = 0.0017$) but not in in the $\Delta mecB$ (increment of 1.39× on A549 relative to $CTH^{-/-}$, $P = 0.107$). This points to MecB as the major responsible for the specific increase in persulfidation in response to host challenge, and therefore conceivably as the most relevant of the persulfidating proteins for fungal virulence. In addition, we measured total levels of persulfidation in hyphae challenged with A549 or $CTH^{-/-}$, using the persulfidation fluorescence imaging protocol [20], and an automated image processing and analysis macro created to mask the fungus from the human cells based on fungal-specific Calcofluor White staining (S9B Fig). As expected, the mean persulfidation fluorescence signal in *A. fumigatus* wild-type hyphae significantly exceeded the $\Delta mecB$ mutant hyphae in all conditions. In agreement with the previous result, the level of persulfidation was significantly higher when *A. fumigatus* was in contact with A549 compared to $CTH^{-/-}$ cells for wild type (increment of 1.24× on A549 relative to $CTH^{-/-}$, $P < 0.0001$) but not for $\Delta mecB$ (increment of 1.06× on A549 relative to $CTH^{-/-}$, $P = 0.89$) hyphae (Fig 4B). Therefore, the level of host persulfidation influences the level of *A. fumigatus* persulfidation. We hypothesized that this interaction may occur through 2 alternative mechanisms, either by host-produced $H_2S$ diffusing into fungal cells and directly impacting the level of persulfidation or by an active response in *A. fumigatus* to the stress caused by the host cells. To gain some further insight, we tested *A. fumigatus* persulfidation levels in vitro in the presence of a sulfide donor or an oxidative stressor. Addition of the $H_2S$ donor GYY4137 did not affect *A. fumigatus* persulfidation levels, while incubation of mycelia in the presence of peroxide increased the persulfidation levels of the wild-type strain by approximately 1.4× ($P = 0.014$), and of the $\Delta mecB$ by approximately 1.3× ($P = 0.052$) (Fig 4C). This suggests that the increase in persulfidation of *A. fumigatus* in the presence of epithelial cells is due to an active response to stress caused by the effector cells, such as oxidative stress, which is partly dependent on the activity of MecB.

Altogether, this agrees with all the results presented and further supports our conclusions: (1) CTH-dependent modulation of persulfidation levels in recipients' lung resident host cells are required for maximum antifungal potency; and (2) a correct, MecB-dependent, level of persulfidation in the fungus is important for its virulence. Future research should aim to understand and unravel the mechanisms that underlie the effects of persulfidation on both the host and the pathogen.

## Discussion

Adaptation is paramount in host–pathogen interactions. Pathogens must be able to adapt to the harsh and varying conditions encountered inside a host. Concomitantly, host cells must respond properly to the challenge to kill the pathogen and mount a proper immune response. Persulfidation is a PTM known to be important for a variety of physiological processes [8]. In pathogens, it has only been studied in *S. aureus*, in which it has been related to the production of virulence factors and cytotoxicity [12,53]. However, the impact of low persulfidation for *S. aureus* virulence in the context of infection was not well defined. In a mouse model of infection, a low persulfidation mutant was reported to cause reduced bacterial burden, but the consequences of low persulfidation for bacterial fitness and/or resistance to host killing were not investigated. Interestingly, a double mutant of the CBS and CTH encoding genes ($\Delta CBS\Delta CTH$) in *S. aureus* had decreased persulfidation and produced supernatants with

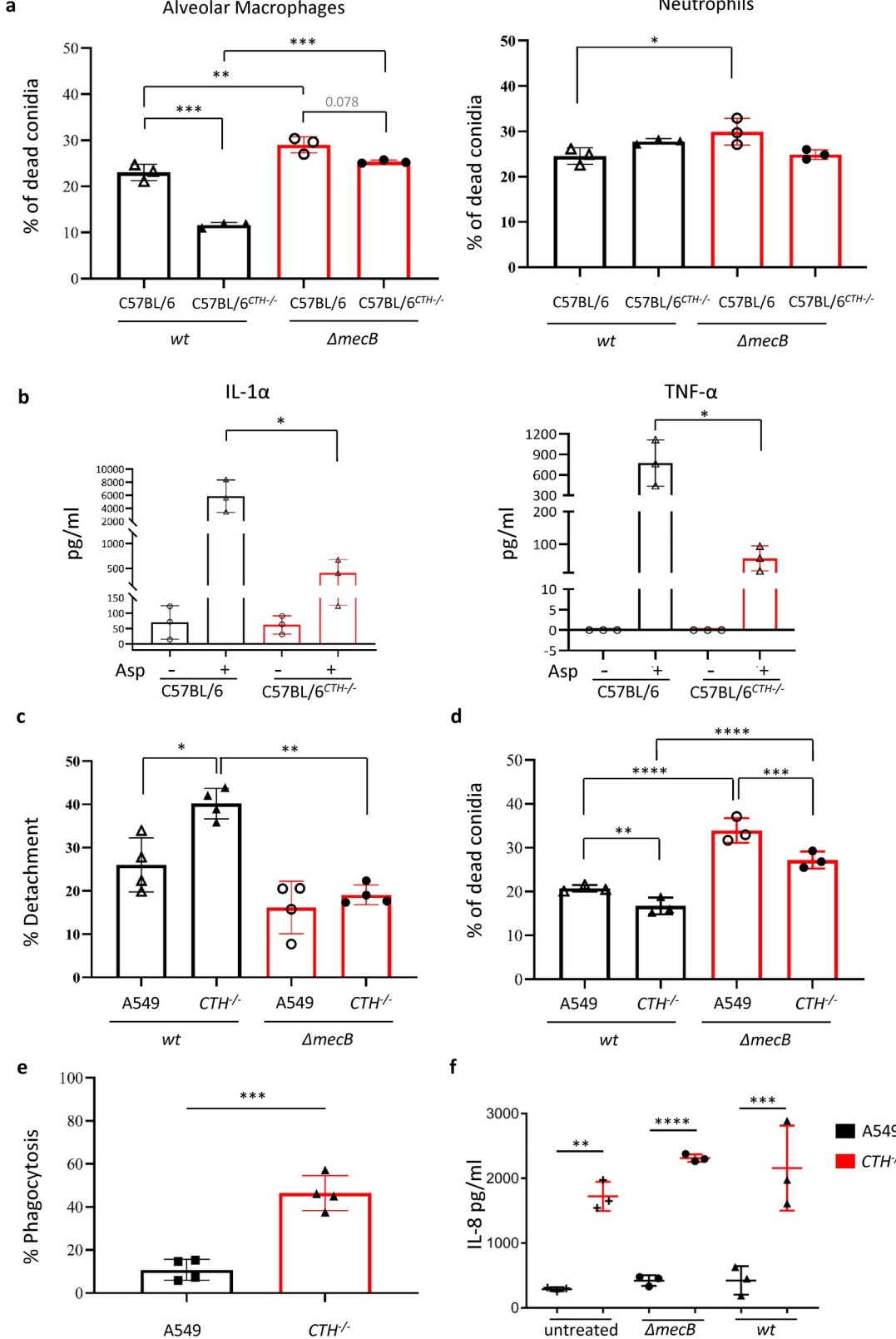

**Fig 3. Defect in CTH reduces the antifungal potency of AMs and alveolar epithelial cells. (a)** AMs isolated from C57BL/6$^{CTH}$ $^{-/-}$ knock-out mice showed a lower capability of killing conidia compared with AMs derived from wt mice, whereas bone

marrow neutrophils killed conidia at the same level. The *A. fumigatus ΔmecB* mutant was more sensitive to killing by both immune cell populations ($n$ = 3 mice, 2 technical replicates; data were analyzed with 2-way ANOVA). **(b)** The level of pro-inflammatory cytokines in the lungs of leukopenic infected mice was lower in C57BL/6$^{CTH−/−}$ knock-out than in wt mice ($n$ = 3 mice, 2 technical replicates; data were analyzed with 1-way ANOVA with Tukey multiple comparisons). **(c)** The $CTH^{−/−}$ alveolar epithelial cell line suffered a higher percentage of detachment than A549 after challenging with *ΔmecB* and wt spores ($n$ = 4, 3 technical replicates; data were analyzed with 2-way ANOVA). **(d)** The $CTH^{−/−}$ alveolar epithelial killed *A. fumigatus* conidia less efficiently compared to its A549 parental line. *A. fumigatus ΔmecB* was killed significantly more than the wt strain ($n$ = 3, 3 technical replicates; data were analyzed with 2-way ANOVA). **(e)** After 4 hours challenge, the $CTH^{−/−}$ cell line internalized significantly more spores than A549 ($n$ = 4 with 4 technical replicates unpaired 2-tailed $t$ test). **(f)** The $CTH^{−/−}$ alveolar epithelial cell line showed a dysregulated increment of IL-8 production ($n$ = 3 with 2 technical replicates, 2-way ANOVA). All data in the figure are depicted as mean ± SD. All numerical values that underlie the data displayed in this figure can be found in S3 Data. AM, alveolar macrophage; CTH, cystathionine γ-lyase; IL, interleukin; SD, standard deviation; TNF-α, tumor necrosis factor alpha; wt, wild-type.

higher cytotoxic potential; however, it did not have reduced virulence [12]. In contrast, we have shown that a partial reduction of persulfidation levels in *A. fumigatus* affects at least 2 enzymatic activities known to be important for virulence, increases susceptibility to host-mediated killing, and ultimately decreases its pathogenic potential. We propose that this reduction in virulence is due to a combined effect of the altered activity of many proteins, affecting a variety of relevant processes (including but not limited to higher susceptibility to killing), the mechanisms of which will need to be determined in future investigations. Furthermore, our results suggest that *mecA* and *mecB* genes are synthetic lethal. We hypothesize that persulfidation may be an essential process for *A. fumigatus*, and these 2 enzymes cannot be eliminated simultaneously because in their absence the levels would decrease below viable levels. We could not validate this hypothesis so far because the TetOFF conditional promoter did not completely switch off gene expression, a limitation that was already reported for other conditional promoters, alcA$^P$ and niiA$^P$ [54]. Therefore, the heterokaryon rescue technique can be considered better suited to validate gene essentiality than conditional promoters and indeed has been used previously on its own to define synthetic lethal genes in *A. fumigatus* [55] and *Aspergillus nidulans* [56]. Synthetic lethality of *mecA* and *mecB* could seem to be in disagreement with *S. aureus*, where a *ΔCBSΔCTH* mutant is viable [12]. However, persulfidation is also not abrogated in this double mutant and was approximately 45% of the wild type, which is similar to the reduction we observed in our single *A. fumigatus ΔmecB* mutant (CTH). Therefore, we speculate that other enzymes (such as MST) or mechanisms may be more relevant for persulfidation in this bacterial pathogen than in *A. fumigatus*. We are currently developing optimized genetic tools which will permit evaluation of our hypothesis that persulfidation is an essential PTM in the future. Interestingly, persulfidation might also be an essential process in mammalian cells, since there is no report of a double *ΔCBSΔCTH* cell line constructed. Therefore, we propose that persulfidation may be an underappreciated essential PTM for all living organisms.

We show that presence of the SNP rs1021737 in the human CTH encoding gene is more common in hematopoietic stem cell transplant recipients that developed IPA than in patients that did not, reflecting an association of the SNP with an overall increased incidence of the disease. This SNP has been proposed to affect phosphorylation sites and decrease protein substrate affinity; therefore, it was expected to reduce the activity of the encoded enzyme [57]. We have shown that indeed the SNP reduces the enzyme's activity to produce H$_2$S from cysteine. As a consequence, human-derived MDMs carrying homozygous rs1021737 are unable to increase persulfidation levels in response to *A. fumigatus* challenge and produce less pro-inflammatory cytokines. Given these defects in myeloid cells, it is surprising that presence of the SNP in the donor compartment does not correlate with a predisposition to develop IPA in transplant recipients. We speculate that lung resident cells (as epithelial cells and AMs) are more important to orchestrate the initial response to fungal challenge and therefore subtle

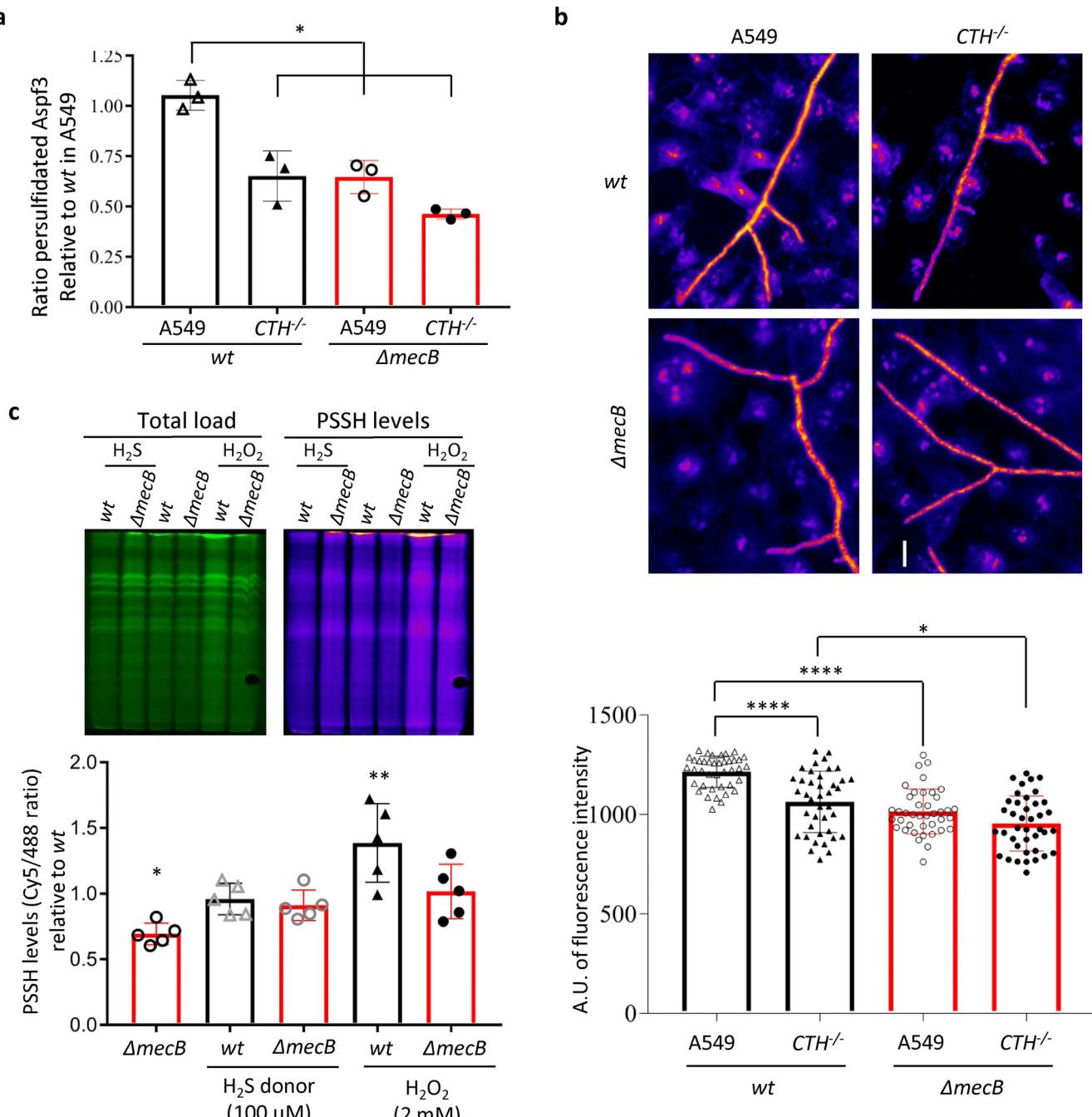

**Fig 4. The level of persulfidation in wt *A. fumigatus* is influenced by contact with the host and by oxidative stress. (a)** The ratio of persulfidated Aspf3 in *A. fumigatus* wild type is higher when the fungus is in contact with A549 than with *CTH*$^{-/-}$ cells. The ratio of persulfidated Aspf3 is always lower in the *A. fumigatus ΔmecB* mutant (*n* = 3). **(b)** Measurement of total persulfidation by fluorescence microscopy reflected that *A. fumigatus* wild type has higher persulfidation level when it is in contact with A549 than with *CTH*$^{-/-}$ cells and that the level of persulfidation is lower in the *A. fumigatus ΔmecB* mutant (*n* = 4 with 2 replicates and 5 photos = hyphae per well). **(c)** The total level of persulfidation did not increase in the presence of a H$_2$S donor. In contrast, incubation with the oxidative stressor H$_2$O$_2$ triggered a significant increment of persulfidation levels in the wt strain, and only a slight increase in the *ΔmecB* mutant (*n* = 5). All data are depicted as mean ± SD and were analyzed using 1-way ANOVA tests. All numerical values that underlie the data displayed in this figure can be found in S4 Data. CTH, cystathionine γ-lyase; H$_2$S, hydrogen sulfide; PSSH, Persulfidation; SD, standard deviation; wt, wild-type.

defects in their activity (caused by the reduced persulfidation level) may trigger the establishment of infection. In contrast, subtle defects in donor cells may not have a big impact on their capacity to fight infection, if the response has been properly orchestrated (notice that CTH-deficient murine neutrophils showed no defect in fungal killing). In agreement with this hypothesis, we demonstrate that lack of CTH enzymatic activity in alveolar epithelial cells reduces the level of persulfidation, which decreases their antifungal potency and deregulates cytokine production and that CTH-deficient murine AMs have a defective fungal killing. The underlying mechanism of these impaired responses could be related to an imbalanced persulfidation of relevant immune regulatory proteins whose activity has been described to be modulated by persulfidation, as NF-κB [41] or tristetraprolin [42]. Therefore, we propose that hematopoietic stem cell transplant recipients carrying homozygous rs1021737 are more prone to develop IPA because of the reduced CTH enzymatic activity and thus persulfidation level of many proteins relevant for the antifungal response of lung resident cells, which leads to a dysregulation of the initial antifungal response, including reduced antifungal potency and an imbalanced cytokine production. Optimal activity of CTH has also been shown to be important for the defense against respiratory syncytial virus in vitro [45] and in a murine model [43], which supports the notion that the action of this enzyme is important for the defense against pathogens. In contrast, it has recently been described that *Mycobacterium tuberculosis* infection is enhanced by CBS [58] and CTH [59] derived $H_2S$ production. Nevertheless, this difference can be explained by *2* conclusions of those studies: (1) *M. tuberculosis* intracellular infection causes an aberrant increase in CBS and CTH transcription in macrophages, which triggers supraphysiological levels of $H_2S$; and (2) such high levels of $H_2S$ are detrimental for macrophage metabolism and beneficial for *M. tuberculosis*. Therefore, it seems to be a *M. tuberculosis*–specific, pathogen-directed mechanism to manipulate the host response for its own benefit. Interestingly, the authors observed cytokine deregulations and impaired antibacterial potency upon imbalanced CTH activity [59], which still agrees with a relevant role of the correct $H_2S$ levels of the host to regulate immune responses. Finally, we propose that many of the effects described on macrophage and bacteria metabolisms are likely exerted by protein persulfidation, something that the authors did not explore.

We have shown that host persulfidation influences the level of persulfidation in *A. fumigatus*. We explored the possibility that a higher production of $H_2S$ in the host could directly diffuse to fungal cells and affect their persulfidation level. However, in contrast to other organisms [12,20,25,60], the sulfide donor GYY4137 alone did not significantly alter persulfidation levels in *A. fumigatus*. We speculate this could be due to a lower diffusion of $H_2S$ through the *A. fumigatus* cell wall and/or a low capacity to oxidize nonenzymatically derived $H_2S$ [7,61]. Alternatively, we hypothesized that the elevated host antifungal potency in competent persulfidating cells may induce more oxidative stress to *A. fumigatus*, which could then be sensed by protein sulfenylation [62] and, in turn, increase persulfidation in the fungal cell [20]. In agreement, we have shown that $H_2O_2$ alone could trigger an increase in persulfidation levels. Therefore, we propose that the persulfidation levels in *A. fumigatus* are modulated as a direct response to host cell defense mechanisms, which are balanced by its own persulfidation levels. Furthermore, our results indicate that sulfide donors could be used to increase persulfidation in lung resident host cells, boosting their antifungal potency, without affecting persulfidation in *A. fumigatus*. Therefore, the use of sulfide donors could be a valuable immunomodulatory strategy to fight *A. fumigatus* infection, as has already been proposed for respiratory syncytial virus infection [63].

In summary, we show that optimal protein persulfidation is important for both *A. fumigatus* pathogenic potential and host antifungal defense and that host persulfidation determines the level of persulfidation in the fungal pathogen. Furthermore, we propose that persulfidation may be an essential cellular process and must be considered as a relevant PTM for infection,

where its modulation may be a promising and novel strategy to target both pathogens and immune responses.

## Materials and methods

### Strains, media, and culture conditions

*E. coli* strain DH5α [64] was used for cloning procedures. Plasmid-carrying *E. coli* strains were routinely grown at 37˚C in Lysogeny Broth LB medium containing 100 μg/ml ampicillin, liquid or solidified with 1.5% agar. All plasmids constructed during this study are listed in **S3 Table** and were generated by using the Seamless Cloning technology (Invitrogen, Altrincham, UK) [19,65].

 *A. fumigatus* strain ATCC 46645 was used as reference recipient. *A. fumigatus* mutants were generated using a standard protoplast protocol [66], and the plasmids are listed in **S3 Table**. Once verified by Southern blot, the deletion cassettes were excised by growing the strains on xylose, as previously described [19,65]. *A. fumigatus* strains were generally cultured in MM (1% glucose, 5 mM tartrate NH$_4$, 4 mM MgSO$_4$, 6 mM KCl, 11 mM KH$_2$PO$_4$, x1 Hutner's trace elements solution; pH 5.5) at 37˚C. To select strains in presence of the resistance marker, 50 μg/ml of hygromycin B or 100 μg/ml of pyrithiamine (InvivoGen, Toulouse, France) were added to the medium.

### Stress sensitivity assays

Stress sensitivity assays were performed by spotting conidial suspensions of $10^4$ conidia down to 1 conidium of each strain in a final volume of 5 μl onto potato dextrose agar (PDA, Oxoid, Hampshire, UK) plates that contained a specific concentration of the appropriate stress treatment. Thermal and hypoxic stress were applied by incubating the plates at 48˚C or 37˚C with 1% O$_2$, respectively. Osmotic stress was studied using 300 mM NaCl and KCl. Cell wall stress was assessed preparing 0.01% SDS, 300 μg/ml Calcofluor White (Sigma-Aldrich, Dorset, UK), 60 μg/ml Congo Red (Sigma-Aldrich), and 7.5 mM Caffeine (Sigma-Aldrich) containing solid medium. Oxidative stress was tested with serial diluted concentrations of H$_2$O$_2$, Diamide, Menadione, and Fludioxonil (Sigma-Aldrich) in 96-well plates inoculated with $10^3$ conidia. Plates were incubated at 37˚C for 48 hours, and the growth of *A. fumigatus* mutant isolates was compared to the wild type.

### Extraction and manipulation of nucleic acids

Recombinant DNA technology was carried out following standard protocols [67]. Phusion high-fidelity DNA polymerase (Thermo Fisher Scientific, Waltham, Massachusetts, USA) was generally used in polymerase chain reactions (PCRs), and results essential cloning steps were verified by Sanger sequencing. Fungal genomic DNA was isolated following the protocol described by Kolar and colleagues [68], and Southern blot analyses were performed as described by Southern [69,70] using the Amersham ECL Direct Labeling and Detection System (GE Healthcare, Chalfont Saint Giles, UK).

 DNA was isolated from murine lungs as follows: Explanted, frozen lungs were lyophilized for 48 hours in a CoolSafe ScanVac freeze drier connected to a VacuuBrand pump and subsequently ground in the presence of liquid nitrogen. DNA was isolated from the powder using the DNeasy Blood & Tissue Kit (Qiagen). DNA concentration and quality were measured using a NanoDrop 2000 (Thermo Fisher Scientific, Erlangen, Germany).

### Dimedone switch method

The detection of persulfidated proteins was assessed following the protocol described by Zivanovic and colleagues [20]. Confluent A549 and *CTH*$^{-/-}$ alveolar epithelial cell lines were lysed

using HEN Buffer (0.5 M EDTA, 20% SDS, 1% NP-40, 10 mM Neocuproine and 0.1M HEPES, pH 7.4) supplemented with 5 mM 4-chloro-7-nitrobenzofurazan (NBF-Cl, Sigma-Aldrich), whereas mycelia from *A. fumigatus* were first ground in the presence of liquid nitrogen and samples then incubated in HEN lysis buffer (50 mM HEPES; 1 mM EDTA, 2% SDS, 1% NP-40, 0.1 mM Neocuproine; pH 7.4) supplemented with 1X cOmplete, Mini EDTA-free Protease Inhibitor Cocktail (Roche, Welwyn Garden City, UK) and 5 mM NBF-Cl at 37°C for 30 minutes. Afterwards, proteins were precipitated twice in the presence of methanol and chloroform (1:1:0.25) at a centrifugation of 14,000 rpm for 15 minutes at 4°C, whereby the pellets were resuspended in 50 mM HEPES (pH 7.4) supplemented with 2% SDS. Once the proteins were completely dissolved, the concentration was determined by BCA assay and adjusted to 2 to 3 mg/ml. Samples were then incubated with 25 μM pre-click mix (1 mM Daz-2, 1 mM Cy5 alkyne, 2 mM TBTA Cu Complex, 4 mM L-arcorbic acid, 30% acetonitrile and 20 mM EDTA in PBS) for 30 minutes at 37°C and precipitated as described above. Subsequently, proteins were resuspended in 50 mM HEPES supplemented with 2% SDS and separated using SDS-PAGE on 12% (w/v) polyacrylamide gels. Gels were fixed in 12.5% methanol and 4% acetic acid for 30 minutes. Persulfidation levels were measured using a Typhoon 9500 Imaging System (GE Healthcare) with Cy5 and Alexa488 fluorescence detection channels. Persulfidation levels were assessed normalizing the intensity of lines in the Cy5 image to the Alexa488 image. Mutant samples were then compared to the levels to the reference samples (*A. fumigatus* wild type or A549 cells).

## Liquid chromatography–mass spectrometry

Samples were process by the Biological Mass Spectrometry Facility in the Faculty of Biology Medicine and Health, University of Manchester. Bands of interest were excised from the acrylamide gel and dehydrated using acetonitrile followed by vacuum centrifugation. Dried gel pieces were reduced with 10 mM dithiothreitol and alkylated with 55 mM iodoacetamide. Gel pieces were then washed alternately with 25 mM ammonium bicarbonate followed by acetonitrile. This was repeated, and the gel pieces dried by vacuum centrifugation. Samples were digested with trypsin overnight at 37°C. Digested samples were analyzed by liquid chromatography–tandem mass spectrometry (LC–MS/MS) using an UltiMate 3000 Rapid Separation LC (RSLC, Dionex, Sunnyvale, California, USA) coupled to an Orbitrap Elite (Thermo Fisher Scientific) mass spectrometer. Peptide mixtures were separated using a gradient from 92% A (0.1% FA in water) and 8% B (0.1% FA in acetonitrile) to 33% B, in 44 minutes at 300 nL min-1, using a 75 mm × 250 μm i.d. 1.7 μM CSH C18, analytical column (Waters, Cedex, France). Peptides were selected for fragmentation automatically by data dependant analysis. Data produced were searched using Mascot (Matrix Science, UK), against the UniProt database with taxonomy of *A. fumigatus* selected. Data were validated using Scaffold (Proteome Software, Portland, USA).

## Immunoblotting

The enrichment of persulfidated proteins was carried out following the Dimedone switch method. Proteins from the total lysates and persulfidated proteins were separated by SDS-PAGE on 12% (w/v) polyacrylamide gels to determine the total and persulfidated Aspf3 protein content, respectively. Afterwards, they were transferred to a polyvinylidene difluoride (PVDF) membrane using the Trans-Blot Turbo Transfer System (Bio-Rad, Watford, UK). The detection of Aspf3 was carried out with a rabbit polyclonal anti-Asp f3 antiserum [29] and anti-rabbit IgG HRP-linked antibody (Cell Signaling Technology, London, UK) as primary and secondary antibody, respectively. Gels were revealed by using SuperSignal West Pico PLUS Chemiluminescent Substrate (Thermo Fisher Scientific) and the ChemiDoc XRS+ Imaging System (Bio-Rad).

To detect monomers and dimers of Aspf3, proteins were isolated from ground *A. fumigatus* mycelia using nonreducing buffer (100 mM Tris-HCl pH 7.5; 0.1% Triton X-100; 5% glycerol; 1mM EDTA) supplemented with 100 mM acetamide (Sigma) and 1X cOmplete, Mini EDTA-free Protease Inhibitor Cocktail (Roche). The proteins were separated by nonreducing SDS-PAGE on 12% (w/v) polyacrylamide gels. Protein transfer and Aspf3 detection was performed exactly as described above.

### Detection of persulfidation by epifluorescence microscopy

The detection of persulfidation proteins by fluorescence microscopy was carried out following a protocol described by Zivanovic and colleagues [20]. Briefly, $10^4$ conidia/ml of each *A. fumigatus* isolate were grown in a μ-slide 8-well culture chamber (IBIDI, Gräfelfing, Germany) with 200 μl of MM or DMEM (Sigma-Aldrich) supplemented with 10% fetal bovine serum (FBS; Sigma-Aldrich) and 1% Penicillin/Streptomycin at 37˚C for 16 hours. A549 and $CTH^{-/-}$ alveolar epithelial cells were seeded on μ-slide 8-well culture chambers and incubated to 60% to 70% confluency. For infections, confluent A549 and $CTH^{-/-}$ cells were challenged with $10^4$ conidia/ml of *A. fumigatus ΔmecB* or wild type for 16 hours. Samples were washed twice with PBS (Sigma-Aldrich) and incubated with 1 mM NBF-Cl for 30 minutes. For infections, Calcofluor White staining (Sigma-Aldrich) was performed for 15 minutes at room temperature (RT). Subsequently, samples were washed with PBS and fixed with cold methanol and acetone at −20˚C for 20 minutes and 5 minutes, respectively. Once fixed, samples were incubated with 1 mM CINF for 1 hour at 37˚C and washed with 30% methanol in PBS twice. Afterwards, they were incubated with 10 μM pre-click mix for 1 hour at 37˚C and washed with methanol twice. Cellular images were captured on a motorized widefield Nikon TE2000-E microscope equipped with a 20× (0.75NA) CFI Plan Apo Lambda objective lens (Nikon, Kingston-Upon-Thames, UK) and a Hamamatsu Orca-ER CCD camera (Hamamatsu Photonics, UK) using MetaMorph v7.7.6.0 software. Fluorescence was captured using a CoolLED *p*E-2 excitation system; a 635-nm LED array module with a Brightline LED-CY5-A-000-ZERO filter (Semrock, New York, USA) for Cy5; a 475-nm LED array module with a Nikon B-2A filter for Alexa488; and a 380-nm LED array module with a Nikon UV-2A for Calcofluor White imaging.

### Automated image analysis

Images (16-bit) were processed and analyzed using the software Fiji [71] http://rsbweb.nih.gov/ij/ and a suite of in-house "Fungal Probe Measure" and "Fungal Infection Probe Measure" macros written in the IJ1 macro language (S1 and S2 Files). Briefly, the Alexa488 channel images were segmented based on local thresholding and mathematical morphology to generate a mask which was used to measure the mean Alexa488 and Cy5 fluorescence intensities in hyphae or epithelial cells. Additionally, CFW channel images were used to automatically segment hyphae in images containing mixed epithelial and fungal cell persulfidation fluorescence (S6 Fig).

### $H_2O_2$ detoxifying activity

The $H_2O_2$ detoxifying activity was measured following 2 different protocols. First, we used a modified protocol published by Nelson and Parsonage [31], which directly monitors the decrease of tert-Butyl hydroperoxide concentration in the presence of Fe(II) and xylenol orange. The enzymatic assay was performed with *A. fumigatus* wild type, *ΔmecB*, and *Δaspf3* as control. Overnight cultures were incubated in the presence of 20 μM $H_2O_2$ to promote the over-expression of the peroxiredoxin at 37˚C for 45 minutes. Subsequently, 200 μM of tert-

Butyl hydroperoxide (Sigma-Aldrich) was added, and its degradation was studied by transferring 20 μl of the culture into a 96-well plate containing 180 μl of FOX working reagent (25 mM ammonium ferrous sulfate in 2.5 M sulfuric acid + 100 mM sorbitol and 125 μM xylenol orange). To determine the concentration of tert-Butyl hydroperoxide in each well, a standard curve up to 200 μM t-butyl hydroperoxide was generated. The absorbance was read at 560 nm, and the concentration of each sample was calculated using the standard curve. The degradation rate of each condition was determined by the slope of the linear regression line obtained in each sample along the first 7 minutes. Finally, the degradation rates of *ΔmecB* and *Δaspf3* were compared to wild type to assess significant differences in the enzymatic activity. This assay was carried out in 4 biological replicates.

We also measured peroxidatic activity of *A. fumigatus* lysates following the protocol described by Kim and colleagues [32]. A total of 2 mM $H_2O_2$ was added to overnight grown *A. fumigatus* mycelia and strains incubated for 2 hours. Mycelia were filtered, snap-frozen, ground in the presence of $N_2$ liquid, and lyophilized for 24 hours in a CoolSafe ScanVac freeze drier. Lyophilized mycelia were resuspended in HEN buffer (50 mM Hepes, 1 mM EDTA, 0.1 mM Neocuproine) buffer pH 7.4 containing 0.1% SDS and 1% protease inhibitors and protein concentration adjusted to 2 mg/mL. Moreover, 200 μM NADPH, 3 μM human recombinant Trx and 1.5 μM rat TrxR were added into each lysate. The reaction was started by adding 100 μM $H_2O_2$ and changes of NADPH absorbance monitored at 340 nm using Agilent (Cedex, France) UV8454 spectrophotometer (at RT). Representative traces of 3 independent measurements originating from 3 biological replicates are shown.

## Labeling of protein sulfenylation

Protein extraction from *A. fumigatus* was performed by adding the cold HEN buffer (50 mM Hepes, 1 mM EDTA, 0.1 mM Neocuproine, 1% IGEPAL and 2% SDS, pH 7.4) supplemented with 100 μM BTD and 1 mM PMSF (Sigma, P7626) to lyophilized mycelia in 200:1 ratio. Samples were vortexed 4 times for 10 seconds with 20-second breaks on ice. Lysates were cleared by centrifugation at 20,000 x g for 15 minutes at +4˚C, supernatant was transferred to a new tube, and methanol/chloroform precipitation was performed. Protein pellets were dissolved in 50 mM HEPES supplemented with 2% SDS and adjusted to 2 mg/ml, followed by incubation with 10 mM ClNBF for 1 hour at 37˚C (protected from light). Methanol/chloroform precipitation was performed, and pellets were dissolved in HEPES + 2% SDS and adjusted to 3 mg/ml for click reaction. Acetonitrile (30% vol/vol), Cyanine5 azide (100 μM final), Copper(II)-TBTA (200 μM final), and in situ prepared ascorbic acid (800 μM) were added sequentially to samples and gently vortexed. Incubation was performed for 2 hours at RT, slowly mixing and protected from light. Reaction was quenched by methanol/chloroform precipitation. Pellets were dissolved in HEPES + 2% SDS, adjusted to 2.5 mg/ml, mixed with Lammeli (4X) buffer (Bio-Rad) supplemented with 10% β-mercaptoethanol, and boiled at 95˚C for 6 minutes. After SDS-PAGE, protein gel was fixed and recorded on Typhoon at 488 and 635 nm for Cy2 and Cy5 signal, respectively.

## Labeling of protein sulfinylation

Protein extraction from *A. fumigatus* was performed by adding the cold HEN buffer (50 mM Hepes, 1 mM EDTA, 0.1 mM Neocuproine, 1% IGEPAL and 2% SDS, pH 7.4) supplemented with 20 mM NEM (Sigma, E3876) and 1 mM PMSF (Sigma, P7626) to lyophilized mycelia in 100:1 ratio. Samples were vortexed 4 times for 10 seconds with 20-second breaks on ice followed by incubation on ice for 10 minutes. Lysates were cleared by centrifugation at 20,000 x g for 15 minutes at +4˚C, supernatant was transferred to a new tube, and methanol/chloroform

precipitation was performed. Protein pellets were dissolved in 50 mM HEPES supplemented with 2% SDS, and protein concentration was determined using DC protein assay (Bio-Rad) and adjusted to approximately 4.5 mg/ml. Samples were then incubated with 100 μM Bio-Dia-Alk [72] for 1 hour at RT, slowly mixing, and the reaction was quenched by methanol/chloroform precipitation. Pellets were dissolved in HEPES supplemented with 0.1% SDS, adjusted to 1 mg/ml and incubated with anti-aspf-3 antibody (2 μl of 10× diluted original stock) for 1 hour at RT, gently mixing. Immunoprecipitation was performed using SureBeads Protein A magnetic beads (Bio-Rad) washed with HEPES+ 0.1% SDS. A total of 50 ul of suspension was added into each sample and left to incubate overnight at + 4˚C. Beads were then washed 3 times with PBST, liquid was carefully removed after the last wash, and 20 μl of 1X buffer (1 equivalent of 4X Lammeli sample buffer supplemented with 10% β-mercaptoethanol was added to 3 equivalents of HEPES + 0.1% SDS) was added. Samples were boiled at 70˚C for 10 minutes, and liquid was transferred to a new tube. Protein samples were resolved by SDS-PAGE and transferred to a nitrocellulose blotting membrane (GE Healthcare). After blocking in 1% BSA in TBST for 1 hour at RT, membrane was washed with TBST and incubated overnight at +4˚C with monoclonal anti-biotin-peroxidase-conjugated antibody (1:1,000, Sigma, A0185). Membrane was visualized using Clarity Western ECL Substrate (Bio-Rad). Aspf-3 levels were determined after an overnight incubation with anti-Aspf-3 antibody at + 4˚C. Goat anti-Rabbit Alexa Fluor 790 secondary antibody (1:20,000, Invitrogen, A11367) was used for antigen detection, and the visualization was performed on Typhoon 5 at 635 nm for Cy5 signal.

Whole protein extracts labeled with Bio-Dia-Alk were additionally used for total sulfynilation levels. Adjusted samples were resolved by SDS-PAGE and transferred to a nitrocellulose membrane followed by blocking in 5% milk for 1 hour at RT. Membrane was washed 5 times for 5 minutes with PBST and incubated with Streptavidin Protein, DyLight 633 (1:4000, Invitrogen, 21844) in PBST for 1 hour at RT. After being well washed with PBST and PBS, respectively, membrane was recorded at 635 nm on Typhoon 5.

## Labeling of peroxiredoxin sulfonylation

For the detection of hyperoxidized peroxiredoxins, proteins were isolated by adding the cold HEN buffer (50 mM Hepes, 1 mM EDTA, 0.1 mM Neocuproine, 1% IGEPAL and 2% SDS, pH 7.4) supplemented with 1 mM PMSF (Sigma, P7626) to lyophilized mycelia in 100:1 ratio. Samples were vortexed 4 times for 10 seconds with 20-second break on ice followed by centrifugation at 20,000 xg for 15 minutes at 4˚C. Supernatant was transferred to a new tube and protein concertation was determined using the DC protein assay. Moreover, 1 equivalent of Lammeli (4X) buffer (Bio-Rad) supplemented with 10% β-mercaptoethanol was added to 3 equivalents of sample and boiled at 95˚C for 6 minutes. Protein samples were resolved by SDS-PAGE and transferred to a nitrocellulose blotting membrane (GE Healthcare). After blocking in 5% BSA in TBS Tween for 1 hour at RT, membrane was incubated overnight at 4˚C with anti-Peroxiredoxin-SO3 (1:2,000, Abcam, ab16830, Berlin, Germany) primary antibody. Goat anti-rabbit IgG (H+L) secondary antibody Cy5 (1:5,000, Invitrogen, A10523) was used for antigen detection, and the visualization was performed on Typhoon model at 635 nm for Cy5 signal.

## Alcohol dehydrogenase activity

Alcohol dehydrogenase activity was measured following a published protocol [73], with minor modifications. Briefly, full *A. fumigatus* proteins were isolated in non-denaturing buffer (50 mM HEPES pH 7.4; 0.1% Triton-X; 1 mM DTT; 1X cOmplete, Mini EDTA-free Protease Inhibitor Cocktail). Protein concentration was measured by BCA assay and adjusted to 50 μg

in 180 μl of reaction buffer (50 mM HEPES pH 7.4; 1 mM DTT; 2 mM MgCl$_2$; 2 mM NADP +). The reaction was initiated by addition of 10 M ethanol, incubated at 37˚C for 30 minutes, where the OD was measured every 5 minutes at 340 nm. The increase in absorbance, due to the conversion of NADP+ to NADPH, was used to measure NADPH concentration based on its extinction coefficient value of 6,220 M$^{-1}$ cm$^{-1}$. The slope of the linear regression of NADPH production over time reflected enzymatic activity. This assay was carried out in biological triplicates.

## Cloning, expression, and purification of recombinant CTH enzymes and H$_2$S production enzymatic assay

Total RNA was isolated from Calu-3 human bronchial epithelial cells using TRIzol (Invitrogen) extraction protocol. cDNA of *CTH* gene (Gene ID 1491) was generated using SuperScript III Reverse Transcriptase (Invitrogen) with random hexamers for first strand amplification and specific primers for gene amplification: Forward JAE668 (5′-CCTGTACTTCCAATCCA ATATGCAGGAAAAAGACGCCTC-3′) and Reverse JAE678 (for wild-type *CTH*) (5′-CGCC GCGATCGCGGATCCTAGCTGTGACTTCCACTTGG-3′) or JAE679 (for SNP *CTH*) (5′-C GCCGCGATCGCGGATCCTAGCTGTGAATTCCACTTGG-3′). PCR products were cloned into pET-His6-TEV-LIC cloning vector (2B-T) (gift from Scott Gradia. Addgene plasmid#29666; http://n2t.net/addgene:29666) linearized with SspI restriction enzyme (New England Biolabs, Hitchin, UK), using GeneArt Seamless Cloning and Assembly kit (Thermo Fisher Scientific) following the manufacturer's instructions. The expression plasmids were transformed into *E. coli* strain Rosetta 2(DE3) (Merck Millipore, Duren, Germany) and expression was carried out with 1 mM IPTG for 72 hours at RT. Soluble protein extracts were purified with Ni-NTA His•Bind Resin (Merck Millipore, Dorset, UK) following the manufacturer's instructions. An enzymatic assay to evaluate H$_2$S production was performed following the protocol described by Chiku and colleagues [74] with minor modifications. Briefly, 2 μg of each enzyme was added to 100 mM HEPES pH 7.4 containing 12 mM cysteine (Sigma) and 4 mM Lead(II) Acetate (Thermo Fisher Scientific). Reactions were incubated at 37˚C, absorbance was measured at 390 nm after 5, 10, 30, and 60 minutes, and the amount of H$_2$S produced for each measurement was calculated with a standard curve (using Na$_2$S –Sigma-). The results are presented as the ratio of H$_2$S produced by the SNP *CTH* with respect to the wild-type enzyme for each condition and time point measured.

## qPCR

SYBR Green JumpStart Taq Ready Mix (Sigma-Aldrich) was used to perform quantitative real-time PCR (qPCR) analyses. To detect the fungal burden, 350 ng of DNA extracted from each infected lung were subjected to qPCR. Primers used to amplify the *A. fumigatus* β-tubulin gene (AFUA_7G00250) were forward, 5′-ACTTCCGCAATGGACGTTAC-3′ and reverse, 5′-GGATGTTGTTGGGAATCCAC-3′. Those designed to amplify the murine actin locus (NM_007393) were forward, 5′-CGAGCACAGCTTCTTTGCAG-3′ and reverse, 5′-CCCATG GTGTCCGTTCTGA-3′. Standard curves were calculated using different concentrations of fungal and murine gDNA pure template. Negative controls containing no template DNA were subjected to the same procedure to exclude or detect any possible contamination. Three technical replicates were prepared for each lung sample. qPCRs were performed using the 7500 Fast Real-Time PCR system (Thermo Fisher Scientific) with the following thermal cycling parameters: 94˚C for 2 minutes and 40 cycles of 94˚C for 15 seconds and 58˚C for 1 minute. Data were analyzed using the 7500 software (Thermo Fisher Scientific). The fungal burden was calculated by normalizing the number of fungal genome equivalents (i.e., number of copies of the tubulin

gene) to the murine genome equivalents in the sample (i.e., number of copies of the actin gene) [75].

## A549 and THP-1 CTH allotype confirmation

Genomic DNA were obtained from A549 or THP-1 cells using the Blood & Cell Culture DNA Midi Kit (Qiagen). The regions of interest were amplified by PCR using the primers JAE690 (5′-AGTGTGGTCTCACTGTGAACTC-3′) and JAE691 (5′-CAAATCTCACCTCCTTCA GAGG-3′) and the Phusion High-Fidelity DNA Polymerase kit (Thermo Fisher), following the manufacturer's instructions. The PCR products were run on a gel and purified with the NucleoSpin Gel & PCR clean-up kit (MACHEREY-NAGEL). Approximately 20 ng of the purified PCR products were sequenced by Sanger Sequencing at the Genomics Technologies Core Facility of the Faculty of Biology, Medicine and Health (University of Manchester). The obtained sequences were aligned to the theoretical CTH genomic sequence using DNADy-namo software (BlueTractorSoftware). Alignments are shown in S9C Fig.

## Disruption of cystathionine γ-lyase in A549 epithelial cell line

The deletion of the CTH from the human pulmonary carcinoma epithelial cell line A549 (American Type Culture Collection, CCL-185) were carried out using Santa Cruz Biotechnology (Dallas, USA) CRISPR/Cas-9, based on the transfection of cells with CTH CRISPR/Cas-9 KO Plasmid (h2) CTH HDR Plasmid (h2).

A549 cells were maintained in DMEM media supplemented with 10% FBS and 1% streptomycin. Cells were seeded at $10^5$ cells per well in a 24-well plate (Cell Star, Greiner Bio-One, Merck, Dorset, UK) and incubated at 37° C, 5% $CO_2$ for 16 hours. Two hours before transfection, medium was removed completely and replaced with serum and antibiotic free DMEM media. Each well was transfected with 0.5 μg of each of the CTH CRISPR/Cas-9 KO (h2) and CTH HDR (h2) plasmids using Lipofectamine 3000 reagent (Thermo Fisher Scientific) according to the manufacturer's instructions. Four hours after transfection, medium was changed by supplemented DMEM. Toxicity controls were only included in standardization experiments. Moreover, 48 hours after transfection, cells were sorted according to manufacturer's instruction and clonal expanded for 2 to 3 weeks. Confirmation of the disruption was carried out by western blot (molecularly demonstration of knock-out genotype). Proteins from cells were extracted using RIPA Lysis Buffer (150 mM NaCl, 1% Nonidet P-40, 0.5% sodium deoxycholate, 0.1% SDS, 25 mM Tris-HCl; pH 7.4) with 1% ß-mercaptoethanol, 2% sodium orthovandate, and 1X cOmplete, Mini EDTA-free Protease Inhibitor Cocktail (Roche). Proteins were recovered by centrifugation at 13,000 rpm for 30 minutes at 4°C, and a concentration of 3 μg/ml of protein was analyzed by western blotting. The detection of *CTH* was carried out with CTH antibody (dilution 1:500, Abcam) and m-IgGκ BP-HRP (1:5,000, Santa Cruz Biotechnology) as primary and secondary antibody, respectively. For normalization of loading, glyceraldehyde 3-phosphate dehydrogenase (GAPDH) was used as housekeeping. In this case, Anti-GAPDH antibody (dilution 1:1,000, Abcam) and HRP Goat Anti-Rabbit IgG (dilution 1:5,000, BD Pharmingen, San Diego, USA) were used as primary and secondary antibodies.

## A549 epithelial monolayer integrity

A549 and $CTH^{-/-}$ cells were maintained at 37°C 5% $CO_2$ in DMEM (Sigma-Aldrich) supplemented with 10% FBS (Sigma-Aldrich) and 1% Penicillin/Streptomycin (Sigma-Aldrich). For all experiments, $10^5$ cells were seeded in 24-well glass bottom plates (Cell Star, Greiner Bio-One) with supplemented DMEM and incubated at 37°C 5% $CO_2$ until confluency. Monolayers were challenged with $10^5$ spores/ml of *A. fumigatus* wild type and *ΔmecB*. Following co-

incubation, monolayers were washed 3 times with PBS, and cells were fixed with 4% formaldehyde for 10 minutes and permeabilized with 0.2% Triton X-100 (Sigma-Aldrich) for 2 minutes. Finally, cells were stained with 300 nM DAPI for 5 minutes. Stained epithelial cells were imaged on the Nikon Te-2000 widefield microscope described above, using a 20× (0.75 NA) Nikon CFI Plan Apo Lambda objective lens and a 380-nm LED array module with a Nikon UV-2A for DAPI imaging. The number of epithelial cells remaining after infection was calculated with an in-house batch image processing and counting algorithm in Fiji [71]. The level of epithelial cell detachment for each condition was calculated relative to the number of cells attached in the negative controls with no fungi. All assays were carried out in biological triplicates.

## Killing assay

Human monocytic THP-1 cells (ATTC, TIB-202) and macrophage-like cell line Raw.264.7 were cultured using RPMI-1640 media and DMEM (Sigma-Aldrich), respectively, supplemented with 10% FBS and 1% Penicillin/Streptomycin at a density of $10^5$ cells/ml. THP-1 cells were differentiated to macrophages using a concentration of 0.1 μM phorbol 12-myristate 13-acetate (PMA, Sigma-Aldrich) for 16 hours. PMA was removed, and cells were incubated for 48 hours at 37°C in RPMI complete medium in order to obtain macrophage-like cells. C57BL/6$^{CTH-/-}$ knock-out mice background were obtained from Prof Isao Ishii (Showa Pharmaceutical University, Tokyo, Japan), and C57BL/6 wild type were purchased from Charles Rivers (Bristol, UK). AMs and bone marrow neutrophils were isolated from C57BL/6 wild-type and C57BL/6$^{CTH-/-}$ mice as follows. Bone marrow neutrophils were flushed from femur and tibia bones with phosphate-buffered saline and isolated using the EasySep Mouse Neutrophil Enrichment Kit (STEMCELL Technologies, Cambridge, UK). AMs were isolated from bronchoalveolar lavage which were obtained by the administration of 1 ml of PBS via intratracheal injection.

The killing assay was carried out following a modified protocol described by Andrianaki and colleagues [76]. Briefly, $10^5$ cells seeded in a 24-well plate (Cell Star, Greiner Bio-One) were challenged with opsonized conidia at a multiplicity of infection of 1:10 and incubated a 37°C with 5% $CO_2$. The opsonization of conidia was carried out by incubating the spores in FBS at 37°C for 30 minutes. One (for Raw.264.7 and THP-1 macrophages) or 2 (for A549 and $CTH^{-/-}$ epithelial cells) hours after conidia challenge, media was removed, and samples were washed 3 times with pre-warm PBS to remove non-phagocytosed conidia. Fresh media was added, and infections were incubated up to 4 hours with macrophages and 6 hours with epithelial cells. AMs and bone marrow neutrophils isolated from C57BL/6 wild-type and C57BL/6$^{CTH-/-}$ knock-out mice were directly incubated up to 6 hours with spores. At the end of the infection, mammalian cells were lysed with Milli-Q water (pH 11) and sonicated to recover intracellular conidia. Finally, conidia were stained with propidium iodide (PI; 10 μg/ml), and the percentage of dead conidia was assessed using the Nikon Te-2000 widefield microscope described above, using a 40× (0.95NA) LED 550 with a Nikon CFI Plan Apo Lambda 40X objective lens and a 550-nm LED array module with Nikon G-2A filter cube. Brightfield images show the total number of conidia, and PI acquired images reveal the number of dead conidia. All assays were carried out in biological triplicates.

## Phagocytosis

To determine differences in *A. fumigatus* spore uptake due to impaired host $CTH^{-/-}$ gene function, $10^5$ A549 and $CTH^{-/-}$ alveolar epithelial cells were seeded in a 24-well plate (Cell Star, Greiner Bio-One) and incubated for 24 hours at 37° C and 5% $CO_2$. Cells were then challenged with $5 \times 10^5$ spores of an *A. fumigatus* strain which constitutively expresses the red

fluorescent protein TdTomato [77], centrifuged for 5 minutes at 100 ×g to synchronize the infection and then incubated for 4 hours at 37˚C, 5% $CO_2$. To stop spore uptake, infected monolayers where then washed 3 times with ice-cold PBS. Labeling of extracellular conidia was performed by incubation with 10 ug/ml Calcofluor White (CFW) solution on PBS for 5 minutes at RT. Cells were then washed 3 times with PBS and fixed with 3% (v/v) formaldehyde for 10 minutes. Experiments were performed in 4 biological and technical replicates.

Automated multi-well imaging 3D fluorescence imaging was performed using a Nikon Ti-U widefield microscope equipped with a Prior H117 automated XYZ stage and Proscan III controller (Prior Scientific, Cambridge, UK). A CoolLED pE-2 excitation system with a 380-nm and 550-nm LED array module and a Brightline LED-DA/FI/TR/CY5-A-000-ZERO multiband filter cube was used to visualize CFW cell wall and cytoplasmic TdTomato fluorescence, respectively. Multi-channel fluorescence images were captured using a 20× (0.75 NA) Nikon CFI Plan Apo Lambda objective lens on a Hamamatsu Flash 4.0 LT+ sCMOS camera (Hamamatsu Photonics) with Nikon Elements acquisition software (v.5.11.01).

Automated phagocytosis analysis was performed using an in-house image processing and analysis script "2D Fungal Phagocytosis Differential" written in Fiji [71] (S3 File), where total TdTomato fluorescing spores and external CFW stained cells were counted to determine the level of internalization occurring. Briefly, images were processed and enumerated based on segmentation and MorphoLibJ library plug-in [78].

### Ethics statement for animal and human research

Animal experimentation was performed under the UK Home Office Project Licence PDF8402B7, and the specific protocols were approved by the University of Manchester Biological Services Facility (BSF).

Approval for the genetic association study was obtained from the Ethics Subcommittee for Life and Health Sciences (SECVS) of the University of Minho (no. 125/014 and no. 014/015), the Ethics Committee for Health of the Instituto Português de Oncologia (IPO)—Porto, Portugal (no. 26/015), the Ethics Committee of the Lisbon Academic Medical Center, Portugal (no. 632/014), and the National Commission for the Protection of Data, Portugal (no. 1950/015). Participants gave written informed consent prior to blood collection.

### Animal infections

Outbred CD1 male mice (Charles Rivers) were used and kept in ICV cages with ad libitum access to food and water. Leukopenic immunosuppression was performed by intraperitoneal injection of 150 mg/kg cyclophosphamide (Baxter, Stockport, UK) on days −3 and −1 plus 1 subcutaneous injection of 250 mg/kg hydrocortisone 21-Acetate (Sigma-Aldrich) on day −1. For the corticosteroid model, mice were injected with a single dose of 40 mg/kg of triamcinolone (Bristol Myers Squibb, Bristol, UK) on day −1. Bacterial infections were prevented by adding 2 g/l neomycin to the drinking water. Mice were anesthetized by exposure to 2% to 3% isoflurane (Sigma-Aldrich) for 5 to 10 minutes and 40 μl of a suspension containing approximately $1.5 \times 10^4$ spores for the leukopenic model (viable counts = $1.5 \times 10^4$ for wild type and $1.45 \times 10^4$ for *ΔmecB*) or approximately $1 \times 10^6$ spores for the corticosteroid model (viable counts = $1.05 \times 10^6$ for wild type and $1.33 \times 10^6$ for *ΔmecB*) was administrated intranasally. Disease progression was monitored twice daily by recording the weight and behavior of the mice. Those individuals showing respiratory distress, hunched posture, poor mobility, or a loss of up to 20% of their body weight were euthanized. Two groups of 5 leukopenic mice were sacrificed at day +3 after infection, and the lungs were harvested and immediately stored in liquid nitrogen for subsequent analysis of fungal burden.

Cytokines in murine lung homogenates was calculated using the BioLegend's LEGENDplex Inflammation panel measured on a Attune NxT Flow Cytometer (Thermo Fisher Scientific) and analyzed using the LEGENDplex Data Analysis Software.

## Patient cohort

A total of 506 hematologic patients undergoing allogeneic hematopoietic stem cell transplantation at the Hospital of Santa Maria, Lisbon and IPO, Porto, between 2009 and 2016 were enrolled in the study. Of these, 483 had available donor and recipient DNA samples and patient-level data. A total of 111 cases of probable/proven IPA were identified according to the 2008 standard criteria from the European Organization for Research and Treatment of Cancer/Mycology Study Group (EORTC/MSG) [79]. Moreover, 21 patients were excluded from the study based on the "possible" classification of infection.

## SNP selection and genotyping

The rs1021737 (S403I) SNP in *CTH* was selected based on previous published evidence of functional consequences to *CTH* function [57,80]. Genotyping of rs1021737 was performed using KASP genotyping assays (LGC Genomics, Hoddesdon, UK) in an Applied Biosystems 7500 Fast Real-Time PCR system (Thermo Fisher Scientific), according to the manufacturer's instructions. Mean call rate for the SNPs was >98%.

## Isolation of PBMCs, generation of MDMs, and cytokine measurements

Peripheral blood mononuclear cells (PBMCs) were enriched from buffy coats or whole blood by density gradient using Histopaque-1077 (Sigma-Aldrich), washed twice in PBS, and resuspended in complete cRPMI (RPMI-1640, 2 mM glutamine, 10% human serum, 10 U/mL penicillin/streptomycin and 10 mM HEPES). Monocytes were isolated from PBMCs by positive magnetic bead separation using anti-CD14+ coated beads (MACS, Miltenyi Biotec, Cologne, Germany), following the manufacturer's instructions. Isolated monocytes were derived into macrophages in cRPMI medium, at a concentration of $1 \times 10^6$ cells/mL in 24-well and 96-well plates (Corning, Flintshire, UK) in the presence of 20 ng/mL recombinant human granulocyte macrophage colony-stimulating or 20 ng/mL of macrophage colony-stimulating factor (GM-CSF or M-CSF, Miltenyi Biotec) for 7 days. The medium was replaced every 3 days, and at the end of incubation, macrophage morphology was confirmed by microscopy using a in a BX61 microscope (Olympus, Hamburg, Germany). MDMs ($5 \times 10^5$/well, in 24-well plates) were infected with *A. fumigatus* conidia at a 1:10 MOI for 24 hours at 37°C and 5% $CO_2$.

Cytokines were quantified using customized Human Premixed Multi-Analyte Kits (R&D Systems, Minneapolis, Minnesota, USA). The cytokine levels on the supernatants of infected MDM were quantified using enzyme-linked immunosorbent assay (ELISA) MAX Deluxe Set kits (BioLegend, London, UK), according to the manufacturer's instructions. All cytokine measurements were performed in duplicates, and concentrations were reported in pg/mL. IL-6 and IL-8 protein levels were measured in cell culture supernatants from A549 and $CTH^{-/-}$ cell lines challenged with *A. fumigatus* using the human IL-6 and IL-8 DuoSet ELISA Development System (R&D Systems) according to manufacturer's instructions.

## Statistical analysis of data

GraphPad Prism (v7.04) was used to analyze and display results. All data were firstly analyzed with the Shapiro–Wilk test and, when the *n* is sufficiently high, the D'Agostino & Pearson normality tests. Normally distributed data were analyzed using unpaired Student *t* test (with

Welch correction when the standard distributions were different) or 1-way ANOVA with Turkey or Dunnett multiple comparison tests. Data which were not normally distributed were analyzed using the Mann–Whitney test or Kruskall–Wallis with Dunn multiple comparisons. Experiments considering 2 variables (*Aspergillus* strain and cell line) were analyzed using a 2-way ANOVA. All graphs in the manuscript depict the mean, and the error bars show the standard deviation (SD). And significance is represented as ****$P < 0.0001$; ***$P < 0.001$; **$P < 0.01$; *$P < 0.05$. For the genetic association study, the frequencies of the *CTH* genotypes were compared among cases of IA and uninfected controls using the Fisher exact t test. A $P$ value $< 0.05$ was considered significant.

## Supporting information

**S1 Fig.** **(a)** Reintroduction of *mecB* gene reconstituted the persulfidation levels, showing that the effect is specific to gene deletion ($n = 3$). All data in are depicted as mean ± SD and were analyzed using 1-way ANOVA with Dunnett multiple comparisons. **(b)** RT-PCRs to measure *mecA* (left panel) and *mecB* (right panel) gene expression relative to transcription in the background single mutant strain ($n = 3$). For both genes, the basal gene expression under the Tet-OFF promoter was much higher than native expression. Addition of Dox strongly reduced transcription, but not to expression levels significantly lower than the native expression of the gene. **(c)** The TetOFF strains grew normally in the presence of Dox. **(d)** Persulfidation levels of the TetOFF strains was similar to that of their single mutant background mutants ($n = 1$). All numerical values that underlie the data displayed in this figure can be found in S5 Data. Dox, doxycycline; RT-PCR, reverse transcription polymerase chain reaction; SD, standard deviation.
(TIF)

**S2 Fig.** **(a)** *A. fumigatus* mutants were not sensitive to high temperature (48˚C), hypoxia, or osmotic stress (300 mM NaCl and KOH). The *ΔmecA* mutant was slightly more sensitive to the cell wall stressor SDS (0.01%), but not Congo Red (30 μg/ml), CalcofluorWhite (200 μg/ml), and Caffeine (5 mM). The phenotype was repeated in 2 independent experiments. **(b)** All mutants were more sensitive to $H_2O_2$ and Fludioxonil and the *ΔmecB* mutant was also more sensitive to Menadione and slightly to Diamide. **(c)** Reconstitution of *mecB* in its natural locus (*mecB+*) restored wt levels of resistance to all oxidative stressors. The sensitivity to oxidative stressors has been assayed in 3 independent experiments. **(d)** The *ΔmecB* mutant grew as the wt on methionine or cysteine (5 mM) as the sole sulfur source, indicating that absence of the CTH is not impactful on sulfur metabolism. The plates were repeated in 2 independent experiments. CTH, cystathionine γ-lyase; wt, wild-type.
(TIF)

**S3 Fig.** (a) Persulfidated proteins were detected by 2 different methods, the biotin-thiol assay (persulfidated proteins are eluted in the presence of DTT) and the Tag-switch. (b) Pathway enrichment analysis of persulfidated proteins using the YeastEnrichr platform showed a significant enrichment of GO terms. (c) Representative western blot of full lysates and persulfidated enriched fraction of wt and ΔmecB protein extracts using an Aspf3 antiserum. All numerical values that underlie the data displayed in this figure can be found in S6 Data. GO, Gene Ontology; wt, wild-type.
(TIF)

**S4 Fig.** **(a)** Degradation rate of tert-Butyl hydroperoxide reflects $H_2O_2$ detoxifying activity. The *Δaspf3* mutant, a control of reduced peroxiredoxin activity, showed a strong reduction in degradation rate. The *ΔmecB* mutant also had a lower degradation rate. The curve shows one

representative experiment with 3 technical replicates. **(b)** The ratio of NADPH oxidation, and consequent decrease in absorbance at 340 nm, reflects peroxiredoxin activity in the thioredoxin-dependent assay. Both $\Delta mecB$ and $\Delta aspf3$ had a strong reduction in enzymatic activity. The curve shows 3 biological replicates. **(c)** Representative nonreducing western blot, used to differentiate between Asp3 monomers and dimers. Two bands of dimer Aspf3 can be observed (not in the $\Delta aspf3$ control), likely because Aspf3 can form homodimers and heterodimers with other peroxiredoxins. Quantification of the ratio of monomers (= oxidized) and dimers (= reduced) showed that the oxidation status of Aspf3 is higher in $\Delta mecB$ ($n = 3$, $P = 0.04$, unpaired $t$ test with Welch correction). **(d)** Representative gel and quantification of PSOH. In the presence of $H_2O_2$, the $\Delta mecB$ mutant showed significantly higher levels of oxidation than the wild type ($n = 2$, wild type+$H_2O_2$ vs. $\Delta mecB$+$H_2O_2$ $P = 0.04$, unpaired 2-tailed $t$ test). **(e)** Total protein fraction was labeled with Biotinylated-Dia-Alk probe (binds $PSO_2H$) and blotted with Streptavidin DyLight 633. In the presence of $H_2O_2$, the level of hyperoxidation (sulfinylation) was clearly higher in the $\Delta mecB$ mutant than in wild type ($n = 1$). **(f)** Total protein fraction was labeled with Biotinylated-Dia-Alk probe and IP with the Anti-Aspf3 antiserum. The IP fraction was blotted with Anti-Aspf3 as IP and load control and with Anti-biotin antibody to detect hyperoxidized (sulfinylated) Aspf3. No hyperoxidized Aspf3 could be detected (representative blot of $n = 2$). All numerical values that underlie the data displayed in this figure can be found in S7 Data. IP, immunoprecipitated; PSOH, protein sulfenylation. (TIF)

**S5 Fig. (a)** The ratio of NADP+ reduction to NADPH reflects alcohol dehydrogenase enzymatic activity and is measured by NADPH absorbance at 340 nm. The $\Delta alcC$ mutant showed a strong reduction in NADPH production. The $\Delta mecB$ mutant had a slightly increased speed in NADPH production. The curve shows one representative experiment with 3 technical replicates. **(b)** The $mecB$+ reconstituted strain was killed by Raw.264.7 at the same levels as the wild type strain ($P = 0.92$, 1-way ANOVA with Tukey multiple comparisons) ($n = 3$ with 3 technical replicates). **(c)** Lungs of leukopenic mice infected with the $\Delta mecB$ mutant showed a decreased fungal burden compared to wild type infected ($P = 0.03$), unpaired 2-tailed $t$ test with Welch correction) ($n = 5$, 3 technical replicates per qPCR). Data are depicted as mean ± SD. All numerical values that underlie the data displayed in this figure can be found in S8 Data. SD, standard deviation. (TIF)

**S6 Fig. In IC mice, levels of pro-inflammatory cytokines were slightly elevated in C57BL/6$^{CTH-/-}$ compared with wt mice, which was significant in IL-6 ($P = 0.028$).** In leukopenic mice, the pro-inflammatory cytokines IL-1α ($P = 0.0005$) and TNFα ($P = 0.0006$) were significantly elevated in C57BL/6$^{CTH-/-}$ compared with wild-type mice ($n = 3$ mice, with 2 technical replicates). All data are displayed as mean ± SD and analyzed using 1-way ANOVA with Tukey multiple comparisons. All numerical values that underlie the data displayed in this figure can be found in S9 Data. CTH, cystathionine γ-lyase; IC, immunocompetent; IL, interleukin; SD, standard deviation; wt, wild-type. (TIF)

**S7 Fig. (a)** Representative western blot of the *CTH* protein in the A549 and its derivative *CTH*$^{-/-}$ epithelial cell lines. GAPDH was used as loading control. **(b)** Representative image of in-gel detection of persulfidation levels in A549 and *CTH*$^{-/-}$ cell lines. NBF-Cl labels persulfides, thiols, sulfenic acids, and amino groups; reaction with amino groups produces the green signal; therefore, it reflects the whole protein context and is used to normalize the persulfidation levels. The red signal is produced by the dimedone-Cy5 labeled probe, which selectively

switches NBF-Cl in persulfide groups [20]. Quantification of persulfidation levels, measured as the level of red signal normalized to the green signal, showed a significant decrease ($P = 0.009$) in persulfidation level of $CTH^{-/-}$ relative to A549 ($n = 4$). Data were analyzed using a 1-sample $t$ test. **(c)** Representative images (scale bar = 15 μm) of persulfidation levels detected by microscopy. All numerical values that underlie the data displayed in this figure can be found in S10 Data. CTH, cystathionine γ-lyase; GAPDH, glyceraldehyde 3-phosphate dehydrogenase; NBF-Cl, 4-chloro-7-nitrobenzofurazan.
(TIF)

**S8 Fig. Representative images of the quantification of phagocytosis.** Red color displays the TdTomato signal, which constitutes the whole conidia population. Green color displays the Calcofluor White signal, which constitutes the extracellular (i.e., non-phagocytosed) conidia population. Scale bar = 30 μm.
(TIF)

**S9 Fig. (a)** Representative western blot of full protein lysate and persulfidated enriched fractions using an Aspf3 antiserum. **(b)** Representative images of the measurement of persulfidation levels in hyphae infecting an epithelial cell monolayer NBF-Cl labels persulfides, thiols, sulfenic acids, and amino groups; reaction with amino groups produces the green signal; therefore, it reflects the whole protein context and is used to normalize the persulfidation levels. The red signal is produced by the dimedone-Cy5 labeled probe, which selectively switches NBF-Cl in persulfide groups [20]. Calcofluor White dye specifically stains chitin in the fungal cell wall. This blue channel can be used to automatically segment hyphae (mask) and hence permits to measure the green and red signals exclusively from fungal cells in images containing mixed fungal and human cells. Scale bar = 30 μm. **(c)** Detail of the alignment of the wt theoretical CTH DNA sequence with the Sanger sequences from A549 and THP-1 PCR-amplified gDNAs. Both cell lines carry the original AGT codon for a serine (S) and therefore do not carry the rs1021737 SNP. CTH, cystathionine γ-lyase; NBF-Cl, 4-chloro-7-nitrobenzofurazan; wt, wild-type.
(TIF)

**S1 Table. The MIC of 3 drugs representative of each type of antifungal (AMB = Ambisome, polyene; VRC = voriconazole, azole; AND = anidulafungin, echinocandin) were calculated for the *A. fumigatus* wt and *ΔmecB* strains.** The range of concentrations tested for each antifungal is depicted in brackets (mg/L). The MIC value of each drug was read at 24 and 48 hours after inoculation, and the calculated values (mg/L) are shown in the table. The assay was run using 2 biological replicates and 2 technical replicates. MIC, minimum inhibitory concentration.
(DOCX)

**S2 Table. List of proteins of the persulfidated enriched fraction identified by mass spectrometry using the *A. fumigatus* Z5 annotation in the Swiss-Prot Trembl database.**
(DOCX)

**S3 Table. Plasmids used in the course of this study.**
(DOCX)

**S1 File. Fiji in-house written macro (in IJ1 macro language) to measure the mean Alexa488 and Cy5 fluorescence intensities in fungal hyphae.** A mask is generated in the Alexa488 channel, which is segmented based on local thresholding and mathematical morphology.
(TXT)

**S2 File. Fiji in-house written macro (in IJ1 macro language) to measure the mean Alexa488 and Cy5 fluorescence intensities specifically in hyphae in images containing mixed epithelial and fungal cells.** A mask is generated in the blue channel (stained with the fungal specific dye CalcoFluor White) to automatically segment fungal hyphae.
(TXT)

**S3 File. Fiji in-house image processing and analysis script to measure phagocytosis.** Total (TdTomato+, red channel) and external (Calcofluor White+, blue channel) spores are counted to determine the level of internalization. Images were processed and enumerated based on segmentation and MorphoLibJ library plug-in.
(TXT)

## Acknowledgments

We would like to thank Ms. Riba Thomas for her magnificent technical support. We thank Dr. Rocio Garcia-Rubio and Dr. Emilia Mellado for assaying the antifungal susceptibility of the $\Delta mecB$ mutant. We would also like to thank Julian Selley and David Knight from the Biological Mass Spectrometry Facility of the Faculty of Biology Medicine and Health (University of Manchester) for the support in mass spectrometry. We acknowledge the use of the Phenotyping Center at Manchester (PCAM) for the use of their microscopes and advanced image analysis workstations and the Genomic Technologies Facility (Faculty of Biology Medicine and Health, University of Manchester) for the Sanger Sequencing. We are grateful to Dr. Isao Ishii for sharing the C57BL/6$^{CTH-/-}$ mice. We also express our gratitude to Dr. Markus Kalkum for kindly providing the anti-Aspf3 serum. We further thank Dalia Sheta for support with the C57BL/6$^{CTH-/-}$ mouse experiment. We are indebted to the members of the IFIGEN Study Group, in particular Dr. António Campos Jr and João F. Lacerda, for their collaboration in the collection of patient material and data. We further thank Dr. Sven Krappmann and Dr. Nir Osherov for critical reading of the manuscript. Help and encouragement from all members of the MFIG are highly appreciated.

## Author Contributions

**Conceptualization:** Paul Bowyer, Agostinho Carvalho, Elaine Bignell, Milos R. Filipovic, Jorge Amich.

**Data curation:** Monica Sueiro-Olivares, Agostinho Carvalho, Milos R. Filipovic, Jorge Amich.

**Formal analysis:** Monica Sueiro-Olivares, Jennifer Scott, Sara Gago, Agostinho Carvalho, Milos R. Filipovic, Jorge Amich.

**Funding acquisition:** Andreas Beilhack, Agostinho Carvalho, Elaine Bignell, Milos R. Filipovic, Jorge Amich.

**Investigation:** Monica Sueiro-Olivares, Jennifer Scott, Sara Gago, Dunja Petrovic, Emilia Kouroussis, Jasmina Zivanovic, Yidong Yu, Marlene Strobel, Cristina Cunha, Rachael Fortune-Grant, Sina Thusek, Jorge Amich.

**Methodology:** Darren Thomson, Milos R. Filipovic, Jorge Amich.

**Project administration:** Elaine Bignell, Jorge Amich.

**Resources:** Andreas Beilhack, Agostinho Carvalho, Milos R. Filipovic.

**Software:** Darren Thomson.

**Supervision:** Andreas Beilhack, Agostinho Carvalho, Milos R. Filipovic, Jorge Amich.

**Validation:** Milos R. Filipovic, Jorge Amich.

**Visualization:** Darren Thomson.

**Writing – original draft:** Monica Sueiro-Olivares, Jorge Amich.

**Writing – review & editing:** Sara Gago, Darren Thomson, Paul Bowyer, Andreas Beilhack, Agostinho Carvalho, Elaine Bignell, Milos R. Filipovic, Jorge Amich.

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
