## [Editor Report · Decision Letter 0]

12 Jun 2020

Dear Dr Amich, 

Thank you very much for submitting your manuscript entitled "Interdependency of host and pathogen protein persulfidation governs disease severity in experimental and human aspergilloses" for consideration as a Research Article by PLOS Biology. Your manuscript was seen by the PLOS editorial staff as well as by an academic editor with relevant expertise. Please accept our apologies for the delay incurred while we sought external advice.

We were interested to read about your exploration of the relationship between protein persulphidation in both A. fumigatus and the mammalian host, and we appreciate the potential implications of your findings for our understanding of the role played by this understudied post-translational modification in the host-pathogen dynamics that underlie aspergillosis. Regretfully, however, we do not feel that your study represents the strength of advance that we require for publication in PLOS Biology.

While we cannot consider your manuscript for publication in PLOS Biology, we suggest that you consider transferring it to PLOS Pathogens. The PLOS journals are editorially independent, so we cannot guarantee it will be reviewed there.

If you would like to transfer your manuscript, as suggested, please click the following link:

If you do NOT wish to transfer your manuscript, please click this link to decline: 

Please note, you can log into the submission sites with the same login that you used to submit to this journal. 

Should you choose to transfer your submission you will receive a confirmation email within 24-48 hours after accepting the transfer. If you have any questions, please feel free to contact the journal at plosbiology@plos.org.

Thank you again for your interest in PLOS Biology. 

Sincerely,

Roli Roberts

Senior Editor

PLOS Biology

---

## [Decision Letter · Decision Letter 1]

23 Aug 2020

Dear Dr Amich,

Thank you very much for submitting your manuscript "Interdependency of host and pathogen protein persulfidation governs disease severity in experimental and human aspergilloses" for consideration as a Discovery Report at PLOS Biology. Your manuscript has been evaluated by the PLOS Biology editors, an Academic Editor with relevant expertise, and by three independent reviewers.

You'll see that the reviewers are broadly positive about your study, but each raises a number of concerns that will need to be addressed by a combination of new experimental data, further analyses, or textual changes.

In light of the reviews (below), we will not be able to accept the current version of the manuscript, but we would welcome re-submission of a much-revised version that takes into account the reviewers' comments. We cannot make any decision about publication until we have seen the revised manuscript and your response to the reviewers' comments. Your revised manuscript is also likely to be sent for further evaluation by the reviewers.

We expect to receive your revised manuscript within 3 months. 

**IMPORTANT - SUBMITTING YOUR REVISION**

*Re-submission Checklist*

*Published Peer Review*

*PLOS Data Policy*

*Blot and Gel Data Policy*

Sincerely,

Roli Roberts

Senior Editor,

rroberts@plos.org,

PLOS Biology

REVIEWERS' COMMENTS:

Reviewer #1: The manuscript by Sueiro-Olivares et al describes roles for the posttranslational modification persulfidation in host-pathogen interactions and virulence of the human fungal pathogen Aspergillus fumigatus. The study tests the importance of both fungal and host persulfidation, by assaying mutants in fungal or mammalian cystatione gamma-lyase. In addition, several A. fumigatus proteins modified by persulfidation have been identified and the peroxiredoxin Aspf3 analysed further in several experiments. The conclusions are that, on the fungal side, persulfidation helps A. fumigatus to survive immune cell attack and drive disease, while on the host side, persulfidation assists immune cell responses (killing of fungal conidia and production of antifungal cytokines). 

The study is interesting and the concept is novel - to my knowledge, this would be the first report of the importance of protein persulfidation in fungal pathogen-host interactions and disease. I have some concerns regarding the experiments and data interpretation, which I outline below. 

* The authors show that the mecB mutant can utilise cysteine or methionine as a sole source of sulfur in vitro on plates (Fig S2d). This result is used to exclude effects of the mecB mutation on cysteine/methionine metabolism and ascribe phenotypes of the mutant exclusively to defective protein persulfidation. I wonder, however, if the metabolic role of mecB has some function in the survival of mutant conidia in macrophages, given the challenging nutritional environment intracellularly in phagocytes. At the very least, some discussion is warranted on this point. 

* The differences in survival between wild type and mecB mutant conidia in response to immune cells are small (for example 28 vs 16% in mouse macrophages and 14 versus 10% in human). I wonder if this small difference can be physiologically relevant. In other words, both wild type and mecB conidia survive to a substantial extent upon immune cell interactions. A similar small effect is seen for the CTH mutation on the ability of alveolar macrophages to kill Aspergillus conidia (Fig 3). The authors should consider discussing these points.

* Western blots of Aspf3 used to determined persulfidated versus total protein should be improved. In particular, the Aspf3 signal in lysates is over-exposed (Fig S3a, S8a) and bands strangely shaped, which would make quantification difficult. It would also be advisable to show full Western blots (as this is supplemental data) and controls for specificity of the antibody.

* Fig S3c: Monomer/dimer quantification of Aspf3: there are multiple or smeared bands where presumably the monomer should be; it is therefore not clear what exactly was quantified. Bands used for quantification could be indicated and full gels (not cropped) shown, with controls for antibody specificity.

* Statistics should be calculated using data points from independent biological repeats of the experiments. It is not clear to me that has been done in all cases. For example, Fig 1f - percentage of dead conidia: n=3 according to the figure legend, but many data points are plotted. I assume these data points are from the 3 technical repeats, 6 photos taken per well, as described in the legend. However, these data points are not all independent of each other - it does not make much sense to do statistics on data obtained from different microscopy fields of the same wells or technical repeats (which control for technical errors, but not for biological reproducibility of the data). Several other figures suffer from the same problem.

Reviewer #2:

Summary

In this paper, authors enquired about the role H2S-mediated cysteine persulfidation has in the adaptation of Aspergillus fumigatus to its host and reciprocally the importance of the H2S host response in the defense against the pathogen. They conclude that persulfidation is important for both the pathogen and the host. 

General comments

The question raised in this study is novel, important, and timely, as new tools are now available to monitor cysteine residues persulfidation. The paper unfortunately suffers from a poor organization and very casual presentation and writing, with many details missing in the text and figure legends, lack of explanations for many of the specific experimental approaches, which makes the reading very difficult and also forces the reader to dig into the methods. Several of the results presented are convincing, in particular le monitoring of persulfidation levels, and killing assays. Other, as described below are not convincing, or lack controls. Overall the conclusions of this study will be important, provided it is better supported by the data and presentation. The title does not reflects what is shown in the paper, with regards to the word interdependency, as not real interdependency is shown, as developed below, and with the wording disease severity, since authors were not able to reproduce a pulmonary aspergillosis in mice, at least with regards to the CTH-/- background (the explanation for their failure is not understandable), and the data from human cannot be taken as statistically significant (this part suffers from a very poor redaction, which makes it very hard to read).

Specific comments, major and minors

1. Failure of the generation of a mecAmecB mutant is interesting, but cannot be taken to say that persulfidation is a vital process in this fungus. This observation might rather be linked to the functions of the transsulfuration pathway in this sulfur assimilating fungus. Authors can attempt deleting one or the other gene with cover of the conditional expression of the other to prove their case

2. Line 125 : what is fludioxonil, and why is it a glutathione-disturbing antifungal, and on which basis this is said?

3. S1b. Aspf3 and Pxr1 are shown to migrate at 25 and 35 kDa, respectively, which is surprising if indeed they represent two peroxiredoxins. What is their theorical size?

4. S3a. Authors should explain the rigor of the quantification of the persulfidation ratio. The only way how to measure this ratio is to compare the anti-Asp3 blot with the persulfidation fluorescence signal. The 3 blots used for this experiment should be shown.

Spore killing assays with macrophages: DmecB more sensitive to killing

5. Lane 159. Peroxiredoxins are usually very abundant enzymes, and it is hard to believe that a 20% decrease in its persulfidation will alter the activity of this enzyme in cells. The assay to measure Asp3 activity is totally non-specific, at least as it is described in the methods, and the differences of activity shown are not really impressive to conclude anything on this activity. Another assay that uses thioredoxin, thioredoxin reductase and NADPH, which specifies the activity of peroxiredoxins as thioredoxin-dependent thiol-based peroxidases should be done. 

6. How persulfidation affects peroxiredoxins is not well understood, if it indeed does anything. This question should first be addressed with recombinant proteins and enzyme assays. Whether persulfidation indeed protects from enzyme sulfinylation has indeed suggested recently, but these data must be confirmed. Anyhow, the blot of S3c cannot be evaluated: at best, it shows an increase in protein abundance, but not the band corresponding to the reduced (hyperoxidized monomer) and disulfide-linked dimer, which should be separated by at least the size of the monomer. How much H2O2 is used in this assay is not indicated. This experiment should be repeated by another method, for instance the use of an ab specific for the hyperoxidized form, or differential Cys residue alkylation.

7. Lane 158 states "in both cases", but which ones?

8. The all paragraph starting at lane 195 should be re-written for the sake of clarity.

9. lane 198: explain what is a balanced persulfidation?

10: fig. 2b: on how many donors of this genotype was the experiment performed. Furthermore, the total lack of persulfidation response cannot intrinsic to the activity of CTH, but rather to some kind of defective signaling. The experiment also shows an highly significant increase in the basal levels of persulfidated proteins in the CTH SNP macrophages, which according to authors should be protective. Authors cannot throw in data without any explanations.

11. Lane 211: the production of inflammatory cytokines comes here out of the blue, without any explanations of the possible link between this process and H2S production.

12. Lane 214: how many individuals were studied? What was the study procedure: the full number of subjects, medical details… should be given? What are the controls in these experiments. A table in the figure is not acceptable, and the figure totally lacks explanations of what is what.

13. Lane 216: the sentence is not understandable: any effect of what?

14. Lanes 239-40: they killed at higher degree compared to what?

15. The paragraph starting at lane 229 should be written again, for the sake of clarity

16. Lane 244: What are these mice: for clarity, start with one class/subject/genetic backround/treatment, genetic, treatment, and then go to the point; then do the other classes. 

17. What this result means?

18. lane 269 what is the value indicated?

19. Lane 292: the hypothesis presented is not clear.

20. Lane 295: what is the meaning of "full"

21. Fig 4a and S8a: as already asked, how is the quantification performed? It should compare the signal obtained by western blot on the precipitated protein, relative to the P-SS fluorescence signal. Furthermore, the protein WB is overloaded to be able to quantify it. It should be reloaded with less protein. The entire gel should be shown. In 4 a, the values for the Wt is 1.1: what this mean?

22. Lane 301: sentence unclear, and explain why it is important.

23. That the level of host persulfidation influences the level of fungal persulfidation is okay, but the experiment proposed fail to answer why. Why the fungus adaptive response should be minored in the CTH-/- mice is not revealed.

Reviewer #3:

This is an interesting, well-written manuscript about a reciprocal relationship of persulfidation between host and pathogen, using in vitro and animal experimental models of aspergillosis, and with clinical associations. A strength of the report is the manner which the problem has been addressed from multiple hypothesis-testing experimental angles.

COMMENTS

1. Abstract "… and predisposes ….". The clinical study about the CTH SNP is a correlation and not a proof of causality (as is correctly summarised on page 4, last paragraph). This should therefore more correctly say " …. and correlates with a predisposition …. "

2. As several in vitro studies rely on studies done using the human cell line THP-1, its CTH allelotype at the SNP location should be explicitly confirmed as wild type. 

Also, although this is less important for the A549 human alveolar macrophage cell line studies as the test group was an intentional CTH knockdown, it would have been prudent also to check their genotype. I note from methods (p27) and Supplementary Figure S4 that the knockout status of A549 was confirmed by Western blot. Yet the line is referred to by its genotype "CTH-/-". Hence, confirm explicitly that the Western blot phenotype confirmed a molecularly-demonstrated knockout genotype.

3. The animal experiments (Fig 1 g-h) appears to have been done once, with n=11 mice per group. Hence displacement between the survival curves in time is highly vulnerable to inequalities in the inoculated dose and delivered inoculated dose of the control and test spore preparation. Were these validated by plating of inoculated volumes? Were these results consistent by replication in experiments using inoculae that were prepared completely independently of each other? At the moment, particularly if the 1g and 1h experiments were performed side-by-side as can't be excluded, there is the possibility that a single unintended difference in the final concentration of two spore suspensions resulting a larger inoculum concentration in the test mutant group could account for these survival curve differences.

4. The statement (line 216-18) dissociates the correlation between IPA incidence and donor SNP genotype, but localises it to the recipient genotype. This prompted me to question whether the alveolar macrophage genotype (presumably donor derived) was relevant, and how the hypothesis/mechanistic picture accommodated this fact. The interpretation/discussion was undisciplined in regard to the distinction between macrophage and alveolar cell genotype (for example line 354-5 focuses on macrophage, non alveolar epithelial cell, genotype in this context). This point is also blurred in lines 33-34 of the abstract ( "correct levels of persulfidation are required for optimal antifungal activity of lung-resident host cells"). I believe that greater clarity regarding what the hypothesis is and what the data show is required on this point.

MINOR COMMENTS

Page 4: " ….. including an estimated 50,000 cases of lethal invasive pulmonary aspergillosis (IPA) (15)." I was interested in these numbers and so looked up reference 15. Actually, it mentions only invasive aspergillosis, not invasive pulmonary aspergillosis, and it does not provide a mortality rate. Although it most case will be pulmonary, and although it will often be lethal, this reference does not establish either of those points.

Page 8: "…..enzymes'…." the apostrophe should precede the "s" as it is only one enzyme being talked about in the possessive

Table S1. The abbreviations in the headings are not defined within the table or elsewhere; while I could guess some, this should not be necessary. What are the bracketed figures in the line R3? What do the numbers within the cells of the table mean?

Lines 331-332 "…the consequences …. was not investigated …." (should be "were").

---

## [Decision Letter · Decision Letter 2]

31 Dec 2020

Dear Jorge,

Thank you very much for submitting a revised version of your manuscript "Fungal and host protein persulfidation are functionally correlated and govern fungal virulence and host antifungal potency" for consideration as a Discovery Report at PLOS Biology. This revised version of your manuscript has been evaluated by the PLOS Biology editors, the Academic Editor and the original reviewers.

You'll see that although reviewer #1 is now satisfied, reviewers #2 and #3 continue to raise some significant concerns. These must be addressed for further consideration, and I should warn you that we're only prepared to consult the reviewers one more time, so you will need to satisfy them. Rev #3 still has ongoing concerns about replication in the in vivo experiments, which must be addressed. Ideally, you should try to address rev #2's second point in a more constructive way than merely removing the Asp f3 sulfinylation work.

In light of the reviews (below), we will not be able to accept the current version of the manuscript, but we would welcome re-submission of a much-revised version that takes into account the reviewers' comments. We cannot make any decision about publication until we have seen the revised manuscript and your response to the reviewers' comments. Your revised manuscript is also likely to be sent for further evaluation by the reviewers.

We expect to receive your revised manuscript within 3 months. 

**IMPORTANT - SUBMITTING YOUR REVISION**

*Re-submission Checklist*

*Published Peer Review*

*PLOS Data Policy*

*Blot and Gel Data Policy*

Best wishes,

Roli

Senior Editor,

rroberts@plos.org,

PLOS Biology

REVIEWS:

Reviewer #1:

My concerns have been adequately addressed.

Reviewer #2:

[identifies himself as Michel Toledano]

Summary

This is a resubmitted manuscript on a host-pathogen reciprocal relationship linking the levels of general persulfidation in A fumigatus and lung epithelial cells, as explored by the use of cells with mutations in CSE, one of the main enzyme for H2S production. The data show that fungi with decreased persulfidation have a decreased pathogenic potential in mice/resistance to killing by macrophages in vitro; reciprocally murine or human cells with a CSE knockout have defect in fungi killing and in mounting an inflammatory response. Further, the data show a positive correlation between the level of persulfidation of host cells and the persulfidation response in fungi. Lastly, authors identify in human databases a SNP in CSE, and show it decreases CSE enzymatic activity (H2S production) and its presence correlates with a higher incidence of invasive aspergillosis in immunocompromised transplant recipients.

General comments 

The efforts of authors in answering and improving the manuscript is fully acknowledged. The criticisms made in the previews review have been handled satisfactorily, with one exception that relates to the part on Asp f3, as indicated below. The paper is much improved, and as previously acknowledged, significant for its general message.

Specific comments

1. The synthetic lethality resulting from MicA and MicB gene deletion should be pursued, because if indeed persulfidation is essential for A fumigatus viability, why is it so should provide a significant clue to the physiological relevance and mode of action of H2S. However, as stated before, there is no proof whatsoever that lethality is the result of decreased H2S production, especially under laboratory conditions with no stress imposition. Therefore, even as a postulate, it is not justified by data.

2. There is still a problem with data pertaining to Aspf f3. That enzyme persulfidation decreases in the MecB mutant is convincing. The problem is again on the activity assays and on the monitoring of Asp f3 sulfinylation. On the assays: the first assay is totally unspecific to Asp f3: authors should thus present the data by saying that inactivating CSE decreases the ability of this fungus to degrade the H2O2 present in the cell culture, without referring to Asp f3, the defective persulfidation of which would cause this scavenging defect. Regarding the other assay, this is again an in vivo assay that requires many biochemical steps through which persulfidation might be lost. Although it shows a decreased activity, these in vivo activity assays are not really reliable: the only way to answer the question raised is by performing in vitro assays with recombinant proteins. On sulfinylation: sensitivity to this PTM is an attribute carried by some, but not all Prx family members, and there is no indication that Asp f3 carries this attribute. In fact, as indicated by the report of Hillmann et al, Asp f3 is homologous to the Prx human variant Prx5 and functionally homologous to the yeast isotype Ahp1, which are enzymes that are both known for not undergoing sulfinylation. Further, the blot of S4D does not show any detectable increase in the sulfinylation signal between untreated and H2O2 exposed Wt cells, which would indicate that this enzyme is not sensitive to hyperoxidation. The suggestion here would be to remove this all part as it significantly depart by its weakness from the quality of the manuscript, especially if it is not in the paper scope.

3. What is the difference between the experiments shown in Fig. 1 h, I and the one intended with CTH-/- mice, but not successful? If it is the same what did it work in the former but not in the latter case?

Reviewer #3:

I have re-read the manuscript and noted the responses to my previous comments. Most are adequately addressed - thankyou for the additional explanations. I have also noted the other reviewer comments and the responses to them.

However, regarding my previous point 3, which questioned the degree of replication of the in vivo experiments, there are still concerns. The authors have indicated that each survival cohort has been run only once, and now revealed that also they were not run concurrently. By providing more information about back-plated determinations of CFU inocula for infectious dose verification, they argue that differences in inoculated doses between groups are not relevant. This scenario is problematic overall because:

(1) the lack of experimental replication means it is not established that the variation due to the test intervention (genotype) is greater than the variation between running the experiment independently on different days (even despite n=11 mice per group). The point being evaluated is the difference in the survival curves between groups; while n=11 within the groups provides some power to discern/resolve differences, when it reflects the course of single cohorts in a single experiment, it does not establish reproducibility.

(2) for the corticosteroid experiment, the stated inocula doses were "1.33×10^6 wild-type and 1.05 ×10^6 ΔmecB", indicating that the WT dose was 30% greater than the mutant dose. Yet following it is stated that "in the corticosteroid experiment the infectious dose of the mutant was higher than the wild-type control, but still caused less mortality", which is not correct based on the doses given, and if the numbers are correct negates the argument being presented in favour of a true mutant-dependent reduction in mortality effect.

(3) The methods section lines (807-809) should provide the verified inocula, given as viable spores (not just spores), as it was established by back-plating (not current revealed in the methods section), and the difference between 1.33 and 1.05 is approx. 31% and should not be stated as being 1 ×10^6 for both groups.

(4) It is not clear if the inocula sizes were almost 100-fold different in the leukopenic and corticosteroid models for an intended reason such as prior knowledge of the appropriate dose in WT animals (if so, provide citation), or if it was the case just by chance, perhaps because the experiments were not done concurrently.

Incidentally, I noted that the competing interest statement in the paper itself (none to declare) is different from that in the front-end material (where one author declares a competing interest).

---

## [Decision Letter · Decision Letter 3]

21 Apr 2021

Dear Jorge,

Thank you for submitting your revised Discovery Report entitled "Fungal and host protein persulfidation are functionally correlated and govern fungal virulence and host antifungal potency" for publication in PLOS Biology. I've now obtained advice from two of the original reviewers and have discussed their comments with the Academic Editor. The Academic Editor also asked me to share some appreciative comments; I've pasted these into the foot of this email. Please accept my apologies for the length of time this has taken.

Based on the reviews, we will probably accept this manuscript for publication, provided you satisfactorily address the following data and other policy-related requests.

IMPORTANT: Please attend to the following:

a) Your current title is somewhat hard to parse; we propose the following modified version: "Fungal and host protein persulfidation are functionally correlated and regulate both virulence and antifungal response," but we're open to other suggestions.

b) Please address my Data Policy requests further down. Essentially you need to supply the numerical values underlying Figs 1ABCDEFGHI, 2AC, 3ABCDEF, 4ABC, S1ABD, S3B, S4ABCD, S5ABC, S6, S7B, and cite the location of the data in the relevant Figure legends.

We expect to receive your revised manuscript within two weeks. 

*Published Peer Review History*

*Early Version*

Sincerely,

Roli

Senior Editor,

rroberts@plos.org,

PLOS Biology

DATA POLICY:

Regardless of the method selected, please ensure that you provide the individual numerical values that underlie the summary data displayed in the following figure panels as they are essential for readers to assess your analysis and to reproduce it: Figs 1ABCDEFGHI, 2AC, 3ABCDEF, 4ABC, S1ABD, S3B, S4ABCD, S5ABC, S6, S7B. NOTE: the numerical data provided should include all replicates AND the way in which the plotted mean and errors were derived (it should not present only the mean/average values).

We require the original, uncropped and minimally adjusted images supporting all blot and gel results reported in an article's figures or Supporting Information files. We will require these files before a manuscript can be accepted so please prepare and upload them now. Please carefully read our guidelines for how to prepare and upload this data: https://journals.plos.org/plosbiology/s/figures#loc-blot-and-gel-reporting-requirements 

DATA NOT SHOWN?

REVIEWERS' COMMENTS:

Reviewer #2:

[identifies himself as Michel B. Toledano]

Authors have satisfactorily answered questions raised in the evaluation.

Reviewer #3:

I agree that the corrected typographical error changes things dramatically. The in vivo results are now consistent with the hypothesis in two independent models of infection, performed non-concurrently. Before this correction, they were not. 

The authors establish what is common practice in the field in the 21 examples cited. Regarding what is compliant with the 3R principles of ethical animal experimentation, generating reproducible data is also an important consideration. Experiments run once are susceptible to once-off errors, as this typo demonstrates. 

In this case, there are now two independent experimental scenarios that both support the author's hypothesis about the virulence difference.

The other points are adequately addressed.

COMMENTS FROM THE ACADEMIC EDITOR:

This looks finalized and ready to accept and publish from my perspective. I appreciate very much the authors responses to reviews, revisions, and careful attention to detail and rigor in finalizing the manuscript. I also appreciate their patience, and their constructive responses to reviews. I think it will be important to explicitly recognize the efforts that the authors have gone to in responding to reviews, and we hope very much that this process has been constructive and helpful in enhancing the quality of the exciting science presented.

---

## [Editor Report · Decision Letter 4]

27 Apr 2021

Dear Jorge,

On behalf of my colleagues and the Academic Editor, Joseph Heitman, I'm pleased to say that we can in principle offer to publish your Discovery Report "Fungal and host protein persulfidation are functionally correlated and modulate both virulence and antifungal response" in PLOS Biology, provided you address any remaining formatting and reporting issues. These will be detailed in an email that will follow this letter and that you will usually receive within 2-3 business days, during which time no action is required from you. Please note that we will not be able to formally accept your manuscript and schedule it for publication until you have made the required changes.

PRESS: We frequently collaborate with press offices. If your institution or institutions have a press office, please notify them about your upcoming paper at this point, to enable them to help maximise its impact. If the press office is planning to promote your findings, we would be grateful if they could coordinate with biologypress@plos.org. If you have not yet opted out of the early version process, we ask that you notify us immediately of any press plans so that we may do so on your behalf.

Thank you again for supporting Open Access publishing. We look forward to publishing your paper in PLOS Biology. 

Best wishes,

Roli 

Roland G Roberts, PhD 

Senior Editor 

PLOS Biology